**EMBO** *reports*

# Progressive chromosome shape changes during cell divisions

Yasutaka Kakui [ID][1], Yoshiharu Kusano [ID][2], Tereza Clarence[3], Maya Lopez [ID][4,5], Todd Fallesen [ID][6], Toru Hirota [ID][2], Bhavin S Khatri [ID][5,7✉] & Frank Uhlmann [ID][4✉]

## Abstract

**Mitotic chromosomes give genome portions the required compaction and mechanical stability for faithful inheritance during cell divisions. They are shaped by the chromosomal condensin complex. Here, we record human chromosome dimensions from their appearance in prophase over successive times in a mitotic arrest. Chromosomes first appear long and uniformly thin. Then, individual chromosome arms become discernible, which continuously shorten and thicken—the longer a chromosome arm, the thicker it becomes. In the search for a molecular explanation of this behavior, given uniform condensin density, the popular *loop extrusion* model provides no obvious means by which longer chromosome arms become thicker. Instead, we find that simulations of an alternative *loop capture* model recapitulate key features of our observations, with re-arranging chromatin rosettes underpinning the gradually developing arm length-to-width relationship. Our analyses portray chromosomes as out-of-equilibrium structures in the process of transitioning towards, but on biologically relevant time scales not typically reaching, steady state.**

**Keywords** Chromosome Formation; Condensin; Polymer Simulations; Loop Extrusion; Loop Capture
**Subject Category** Cell Cycle

## Introduction

The process by which mitotic chromosomes arise in preparation for cell divisions, from apparently amorphous interphase chromatin, has captivated cell biologists for a long time (Flemming, 1882; Sumner, 2003). Chromosomes bestow genome portions the required compaction and mechanical stability for segregation by mitotic spindle forces (Hudson et al, 2003; Poirier and Marko, 2002). While on the one hand constituting stable entities, chromosomes are also known to change their appearance over time. Early observations in plant root cells, in which cell divisions were arrested by the spindle poison colchicine (c-mitosis), revealed how sister arms gradually resolve to yield archetypal X-shaped chromosomes (Levan, 1938; Molè-Bajer, 1958). Successive shape changes have also been reported more recently (Gibcus et al, 2018; Mora-Bermúdez et al, 2006; Shintomi et al, 2017). When chromosomes first become discernible in prophase, they appear uniformly thin (Booth et al, 2016). At later mitotic stages, chromosome arms have shortened and thickened, with longer arms now wider than shorter arms (Kakui et al, 2022). Despite the accumulated evidence for chromosome plasticity, a systematic analysis of chromosome shape changes over time, and an exploration of what these shape changes reveal about chromosome architecture, remains to be performed.

The chromosomal multisubunit protein complex condensin, a member of the Structural Maintenance of Chromosomes (SMC) family, lies at the core of mitotic chromosome formation. No chromosomes form in mitosis without condensin (Hirano, 2016; Uhlmann, 2016). Condensin introduces a layer of mitosis-specific, long-range chromatin contacts. The span of these condensin-mediated chromatin contacts differs between species. Within small budding yeast chromosomes, they span tens of kilobases. In fission yeast, condensin-dependent mitotic interactions reach hundreds of kilobases, whereas they span megabases in the case of human chromosomes (Gibcus et al, 2018; Kakui et al, 2017; Kakui et al, 2022; Lazar-Stefanita et al, 2017; Schalbetter et al, 2017). While the reach of condensin interactions therefore scales with chromosome size amongst organisms, within each species, the contact range is invariant amongst chromosomes of different lengths (Kakui et al, 2022). The molecular mechanism by which condensin establishes mitosis-specific chromatin contacts, and how its species-appropriate contact range is defined, remains incompletely understood (Kakui and Uhlmann, 2018; Kinoshita and Hirano, 2017; Kschonsak and Haering, 2015; Paulson et al, 2021; Yatskevich et al, 2019).

The condensin complex is built around an ABC-ATPase module. ATP binding is required for condensin association with chromosomes, while ATP hydrolysis is necessary to achieve chromosome compaction (Hudson et al, 2008; Kinoshita et al, 2015; Thadani et al, 2018). A popular model posits that the

[1]Waseda Institute for Advanced Study, Waseda University, Tokyo 169-0051, Japan. [2]Division of Experimental Pathology, Cancer Institute of the Japanese Foundation for Cancer Research, Tokyo 135-8550, Japan. [3]Icahn School of Medicine at Mount Sinai, New York, NY 10029, USA. [4]Chromosome Segregation Laboratory, The Francis Crick Institute, London NW1 1AT, UK. [5]Department of Life Sciences, Imperial College London, Silwood Park Campus, Ascot SL5 7PY, UK. [6]Advanced Light Microscopy Science Technology Platform, The Francis Crick Institute, London NW1 1AT, UK. [7]Mechanobiology and Biophysics Laboratory, The Francis Crick Institute, London NW1 1AT, UK.
✉E-mail: b.khatri@imperial.ac.uk; frank.uhlmann@crick.ac.uk

condensin ATPase promotes active extrusion of a chromatin loop, until neighboring condensins meet (Ganji et al, 2018; Goloborodko et al, 2016; Samejima et al, 2025). This 'loop extrusion' model predicts the formation of a central condensin axis from which chromatin loops emerge to create a 'bottlebrush'-like structure. The condensin density and its contact spans have been measured to be equal along short and long chromosome arms (Kakui et al, 2022). The loop-extrusion model therefore predicts that chromatin loops of a similar size will form, so that short and long chromosomes consist of bottle brushes of the same diameter. Thus, short and long chromosome arms would be of the same width.

In an alternative "loop capture" model for chromosome formation, condensin forms chromatin interactions by sequentially topologically entrapping two DNAs that find each other by Brownian diffusion (Cheng et al, 2015; Tang et al, 2023). Simulations of the loop capture mechanism have recapitulated several native-like chromosome features (Forte et al, 2024; Gerguri et al, 2021). Amongst these features, the model predicts formation of rosette-like chromatin interaction foci, consistent with the experimental observation of punctate condensin clusters inside chromosomes (Beckwith et al, 2025; Gerguri et al, 2021; Walther et al, 2018). Dynamic rearrangement of such chromatin rosettes could allow longer chromosome arms to become wider. However, the predicted dimensions of chromosomes formed by a loop capture mechanism, and how these dimensions might change over time, have not yet been investigated.

Here, we analyze the shape changes of human chromosomes, from their appearance in prophase over sequential times in a mitotic arrest, paying special attention to the developing length-to-width relationship. We then compare our measurements to computational loop capture simulations. Initially, uniformly thin chromosomes, or simulated chains, continuously shorten and thicken. The length-to-width relationship in both cases can be approximated by power laws with an exponent that increases over time. Short chromosome arms remain thin, while longer arms become progressively thicker but do not reach a steady state during the times of our observations, or simulations. These considerations put loop capture by condensin forward as a plausible model for chromosome formation.

# Results

## Progressive chromosome shape changes

We synchronized HeLa Kyoto cells at the G2/M boundary by treatment with the cyclin-dependent kinase 1 (CDK1) inhibitor RO-3306. Cells were released from synchronization into medium containing colcemid, allowing mitotic entry but blocking anaphase onset and mitotic exit. Samples were taken at sequential time intervals and processed for chromosome spreading and visualization using the DNA dye 4′,6-diamidino-2-phenylindole (DAPI). Long, thin chromosomes became apparent 12 min after G2/M release, which shortened and thickened by 20 min (Fig. 1). At 30 min, sister chromatids and centromere constrictions became discernible. Sister arms remained cohered while they continued to shorten and thicken up to the 60-minute mark. By 120 min, sister arms separated, remaining connected only at the centromeres, resulting in prototypical x-shaped chromosomes. Over subsequent

hourly intervals, chromosome arms continued to shorten and thicken until we terminated the experiment at 360 min. Our observations suggest that, in mitotically arrested cells, chromosomes undergo continuous shape changes.

To determine when chromosome segregation occurs during unperturbed mitosis, we repeated the above G2/M synchronization experiment but released cells from the CDK1 inhibitor block into medium without colcemid. We followed mitotic progression using the live cell SiR-DNA stain (Fig. EV1A). For the first approximately 30 min, chromosome formation followed a similar trajectory to what we observed in the fixed samples from colcemid-treated cells. Then, unlike in colcemid-arrested cells, chromosomes aligned on a metaphase plate, split and segregated, followed by exit from mitosis. Anaphase onset occurred at $42 \pm 8$ min (mean ± s.d., $n = 50$) following CDK1 inhibitor release. Thus, chromosome segregation usually sets in at a time when sister chromatid arms have individualized but remain cohered.

Above, we observed chromosome shape changes over 360 min in colcemid-arrested cells, a period that is far longer than the usual duration of mitosis. Despite the extended mitotic arrest, 85% of cells remained viable, and they successfully completed cell division when colcemid was eventually washed out at the end of the experiment (Fig. EV1B). Cell divisions therefore do not typically involve the full extent of chromosome shape changes that we here investigate. Nevertheless, these chromosome transformations remain compatible with chromosome segregation and the production of viable progeny.

## Measuring chromosome arm lengths and widths

Chromosomes are biological objects that do not adhere to simple geometric forms. To nevertheless approximate their dimensions, we applied the following tools to chromosome images from our mitotic time course experiment. At the two early time points (12 and 20 min), we manually traced the lengths of each chromosome (Fig. 2A). To determine chromosome widths, we selected straight chromosome regions to which we applied a moving Gaussian fit and derived the average full width at half maximum (Kakui et al, 2022). As centromeres and sister chromatids are not yet discernible, we make the approximation that each of a chromosome's four arms occupies half the length and half the width of a chromosome at these two early time points. We are aware that this approximation overestimates the length of the shorter p-arms and underestimates the length of the longer chromosome q-arms. However, as we will see below, chromosome width is constant amongst short and long chromosome arms at this time, making knowledge of the exact arm lengths less important.

A concern arises from the fixation and spreading protocol that we apply before we measure chromosome dimensions. To investigate whether our treatment distorted chromosome appearance, we compared fixed and spread chromosomes with chromosomes visualized in live cells using the SiR-DNA stain. While the full length of individual chromosomes is harder to trace in the crowded environment of live cells, the measured widths were comparable between live and fixed chromosomes, at the same times following release from G2/M synchronization (Fig. EV1C). We note that quantitatively comparable chromosome widths were also recorded in human prophase using a different fixation protocol and serial block-face scanning electron microscopy (Booth et al, 2016).

  

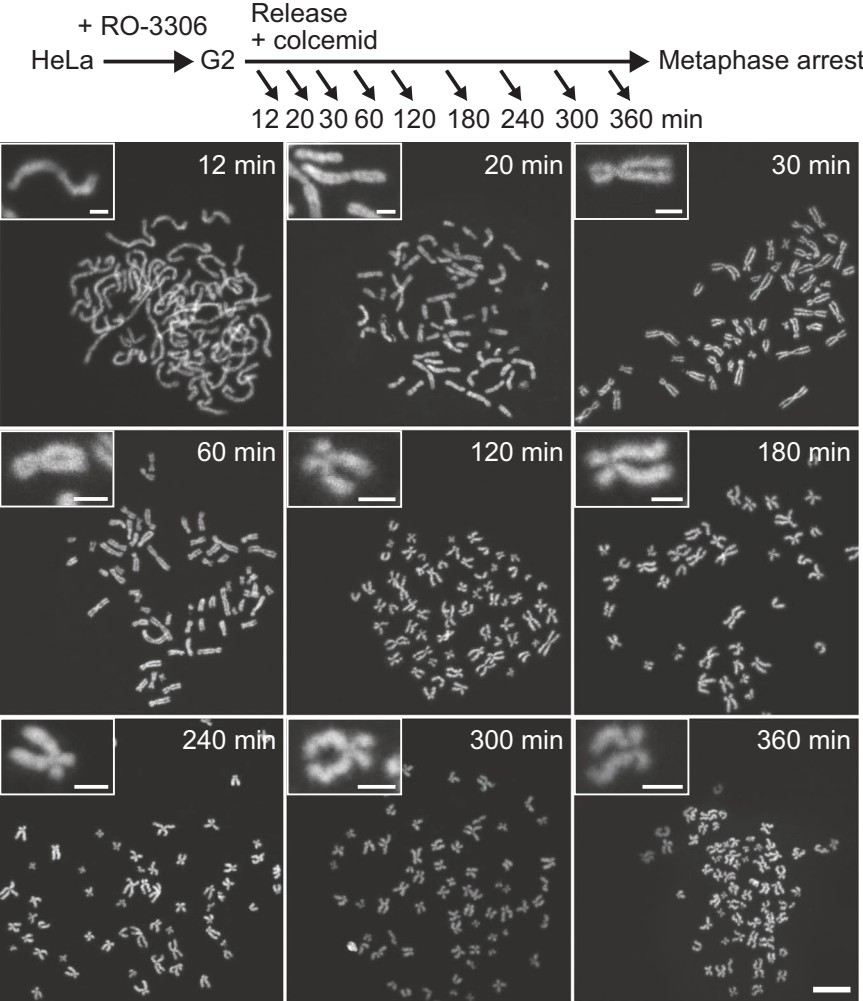

**Figure 1. Experimental design and time series of chromosome shape changes.**

HeLa Kyoto cells were synchronized in G2, released, and maintained in a colcemid-induced mitotic arrest for 360 min. Cells were fixed and chromosome spreads were prepared and stained with the DNA dye 4′,6-diamidino-2-phenylindole (DAPI) at the indicated times. The large images are to scale (scale bar, 10 µm), while the insets show example chromosomes over time at increasing enlargement (scale bars, 2 µm).

Our measurements of fixed chromosomes are therefore representative of chromosomes in their biological context.

From 30 to 360 min, individual chromosome arms are visible, and we traced them using a manual threshold, emulating the half maximum intensity threshold used to determine chromosome width at the two earlier times. To measure arm dimensions, we computationally fitted ellipsoids with the same area as the traced region, then recorded ellipsoid lengths and widths (Fig. 2A). To comprehensively sample chromosome behavior, we measured all chromosome arms from two cells at each time point and aggregated the measurements.

Alongside ellipsoid fitting, we also applied the Gaussian fitting method from the earlier timepoints. We manually traced the length of chromosome arms at 30–360 min, then selected straight sections to which we applied a moving Gaussian fit. This approach gives a more local measure of arm width, at those places that we can access, but it could not be applied to all chromosome arms when they were curved or when sister chromatids were insufficiently separated.

This second approach recapitulated the broad trends observed using ellipsoid sampling, with differences discussed in Fig. EV2.

In the following, we analyze the chromosome dimensions obtained from ellipsoid fits. We emphasize that ellipsoid fitting is a simplification that will sometimes overestimate width, or underestimate length. Our records are not, therefore, meant as an accurate absolute measure of each individual arm. The choice for this method is motivated by its ability to comprehensively measure all chromosome arms, and by the observation that especially shorter arms visually resemble ellipsoids at later times. New developments for observing chromosome behavior should complement our current approach (Stamatov et al, 2025).

## Chromosome arm lengths and widths over time

Plotting chromosome arm lengths and widths over time documents progressive chromosome arm shortening and thickening (Fig. 2B). At early times (12 and 20 min), width was uniform and

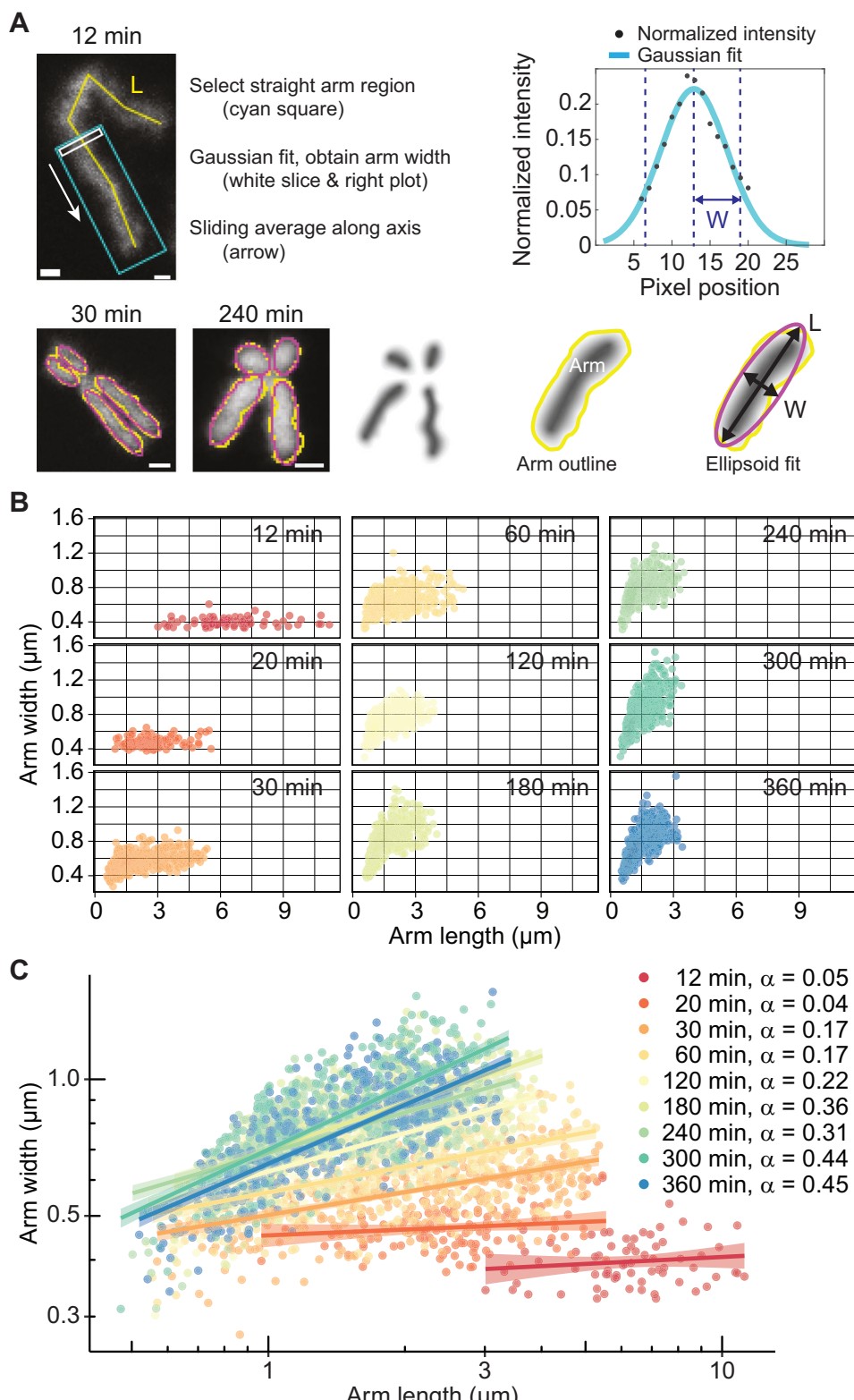

**Figure 2. The chromosome length-to-width ratio reveals a power law relationship with an increasing exponent.**

(A) Illustration of the approaches used to measure chromosome arm lengths and widths at early (12 and 20 min, top) and later (30 to 360 min, bottom) time points (scale bars, 1 μm). (B) Chromosome arm widths as a function of arm length. All chromosome arms were measured and are aggregated from two cells at the indicated times. (C) Arm lengths and width at all above time points plotted on a double logarithmic scale, with power law fits and their 95% confidence intervals, as well as the power law exponents, indicated.

independent of length. As soon as individual chromosome arms become discernible (30 min), longer chromosome arms appeared wider, a trend that became more pronounced as time progressed.

We previously described the mitotic chromosome arm length-to-width relationship at the 30-min time point using a power law $w = c \cdot L^\alpha$ (where $w$ is the chromosome arm width, $L$ is the length, c is a constant and $\alpha$ the power law exponent) (Kakui et al, 2022). To examine whether power laws describe the length-to-width relationship across the time series, we plotted our measurements on a double logarithmic scale (Fig. 2C). A linear distribution of measurements on this scale is indicative of a power law relationship, with the slope of a linear fit reflecting the power law exponent. At early times, when width is invariant between chromosomes, the power law exponent is essentially zero. At successive times, the slope, and thus the power law exponent, increased until reaching 0.45. Below, we will further analyze this developing length-to-width relationship.

Note that a proper test of a power law relationship requires both quantities, here chromosome arm length and width, to vary over several orders of magnitude. This requirement is unavailable in our experiment, so here we use the power law merely as a convenient mathematical form that captures the empirically observed behavior.

## Short arms reach steady state faster

Next, we plotted chromosome arm lengths as a function of time. As our fixation protocol does not allow us to track the progression of individual arms, we instead stratified all measured arms by their length at each time point. We then record the length of every 10th percentile, i.e., the length of the 10th, 20th, etc., until 90th percent longest arm at each time. This method enabled us to follow the behavior of these percentile lengths over time (Fig. 3A). The approach revealed that the shortest percentile arms rapidly shortened within the first 30 min, and after that maintained a relatively constant length. In comparison, the longest percentile chromosome arms showed a different behavior. Shortening also began at a rapid initial rate until 30 min, but then gradually continued until the end of our time course at 360 min. Intermediate percentile arm lengths show a behavior intermediate between these extremes. Because of the uncertainty around actual chromosome arm lengths in the first two time points (which in the absence of a centromere constriction, we approximated as ½ the chromosome length), we did not quantitatively fit a mathematical description to the observed time-dependent length changes. In addition, we cannot be certain of the final steady state length of the longest arms as we terminated the experiment after 360 min. Nevertheless, it becomes qualitatively apparent that short chromosome arms reach steady state length relatively quickly, and before the time when anaphase normally occurs at around 42 min. Longer arms, in contrast, are still on the way to steady state by the time of anaphase onset, and even at the end of our time course experiment.

## A final chromosome roundness

Having realized that shorter chromosome arms reach a final steady state in our time series, we wanted to characterize the dimensions of this state. We therefore looked for a way to distinguish short arms that have reached steady state from longer arms that have not. For that purpose, we plotted chromosome arm roundness over time, where we define roundness $r$ as the ratio of arm width $w$ divided by arm length $L$. Starting from a very small value of ~0.03

at 12 min, roundness gradually increased and spread out over time, reaching an average of ~0.55 at late times (Fig. 3B). Overlaying roundness onto our length-to-width relationship plots confirms that shorter arms generally reach greater roundness, and sooner (Fig. EV3). We then used the final median observed roundness at 360 min of $r = 0.55$ to divide our chromosome arm distribution. Plotting all chromosome arms with a roundness of greater than 0.55 shows that their length-to-width relationship now follows a power law with an exponent of ~0.75, irrespective of the time when these chromosome arms were encountered in our measurements (Fig. 3C). Thus, the final chromosome arm length-to-width steady state can be described by this power law exponent.

On the other hand, chromosome arms with a roundness ≤0.55, as seen before, display a length-to-width relationship with an increasing power law exponent over time (Fig. 3C). This observation confirms that longer chromosome arms continue to change shape until the end of our observation period, with the power law exponent of their length-to-width relationship approaching, but not reaching, that of short chromosome arms that have entered steady state.

## Dimensions of a theoretical and a simulated polymer

To explore a possible underlying mechanism for the observed chromosome arm length and width progression, we turned to polymer simulations. We previously used a biophysical model of a chromatin chain to explore how loop extrusion or loop capture interactions differentially affect simulated chromosome properties (Gerguri et al, 2021), and we now repurpose this model to study the resultant chromosome dimensions. We use a coarse-grained chromatin chain consisting of beads, each modeled to represent a ~2 kb region encompassing ~10 nucleosomes. Beads are connected by springs, with Brownian dynamics determining the stochastic forces on every bead. A soft repulsion term is applied when beads overlap (Fig. 4A). We first investigated whether this representation of a self-avoiding Rouse polymer chain adopts shapes that display theoretically expected behavior and dimensions.

Figure 4B shows representative snapshots of the simulated conformations for polymers of increasing chain lengths (increasing bead numbers, $N$), whilst in equilibrium. The conformations are non-compact and grow in size with increasing $N$. Polymer theory predicts that a random polymer with excluded volume grows in size as $N^\nu$ ($\nu = 0.588$ (Le Guillou and Zinn-Justin, 1977) This power law relationship links *chain length* to *polymer size*. It is different from the previously discussed power law relationship between *chromosome length* and *width*, for which we use the exponent $\alpha$). When we plot the average measured steady state length and width of our simulated chromatin chains as a function of chain length $N$ (Fig. 4C), we see that both measures follow these theoretical expectations closely.

Next, we record the length-to-width relationship of the simulated chains. Because of the unpredictability of random walks in each of the three available dimensions, the resulting polymer shapes are never spherical, but they adopt an ellipsoid shape. Based on probability theory alone, the length, width, and depth of a random polymer should be in proportion length:width:depth = 3.44:1.64:1 (Rudnick and Gaspari, 1987). If we define the "roundness" of the theoretical polymer as its width divided by its length, in analogy to our experimental chromosome arm measurements, the expected polymer roundness based on these proportions is $r = 0.48$. When we now measure the roundness of our simulated chromatin

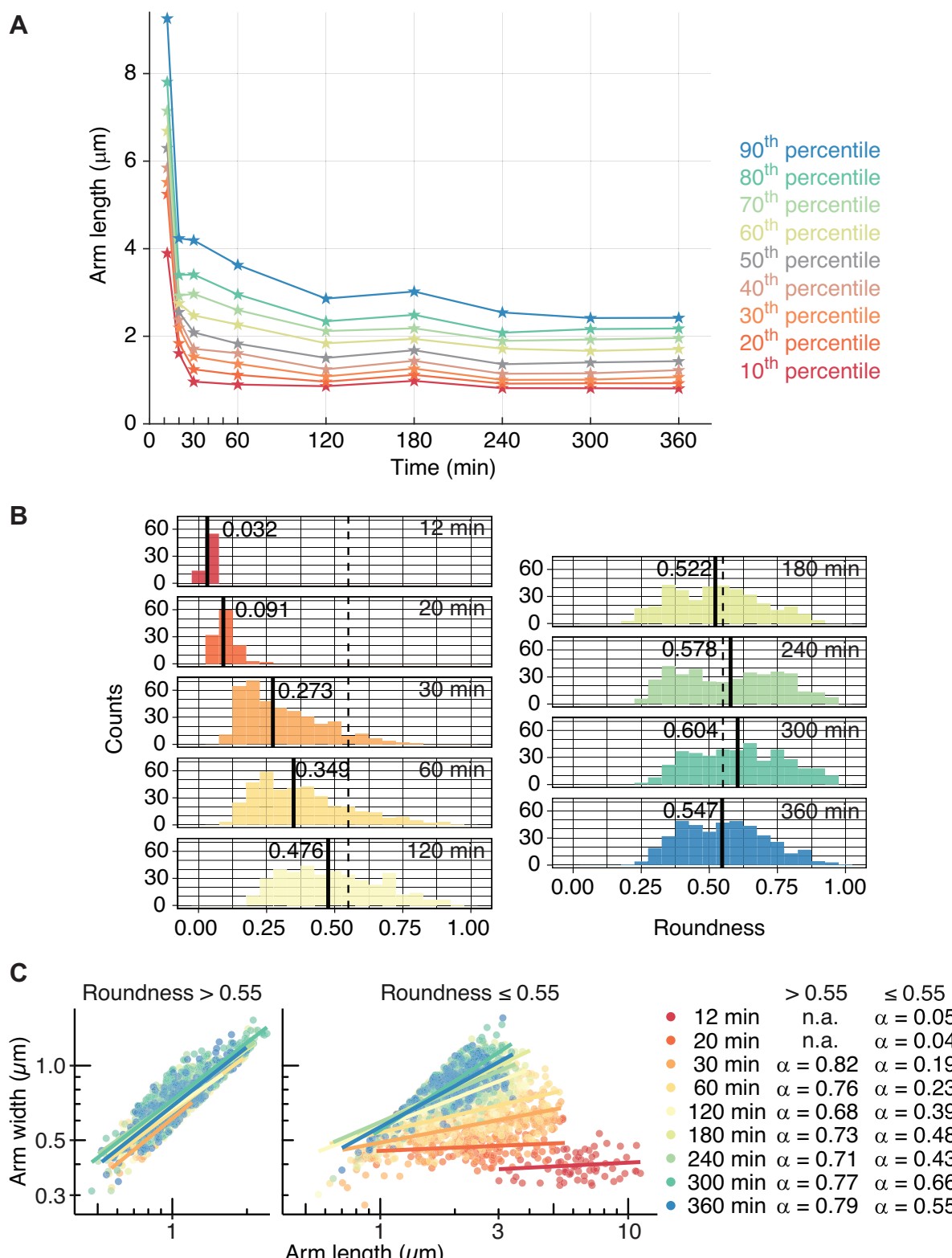

**Figure 3. Short chromosome arms reach steady state faster than long arms.**

(A) Each 10th percentile of recorded chromosome arm lengths was plotted over time. (B) Chromosome arm roundness (width divided by length) as a function of time. Median roundness at each time is indicated by a solid black line. A roundness of 0.55 is highlighted by a dashed line, corresponding to the median roundness at the 360 min time point. (C) Chromosome arm widths as a function of arm length over time, separated into arms with roundness greater vs. equal or smaller than 0.55. Power laws were fitted to both sets and power law exponents are indicated. n.a., not applicable, no arms with roundness greater 0.55 were found at the first two time points.

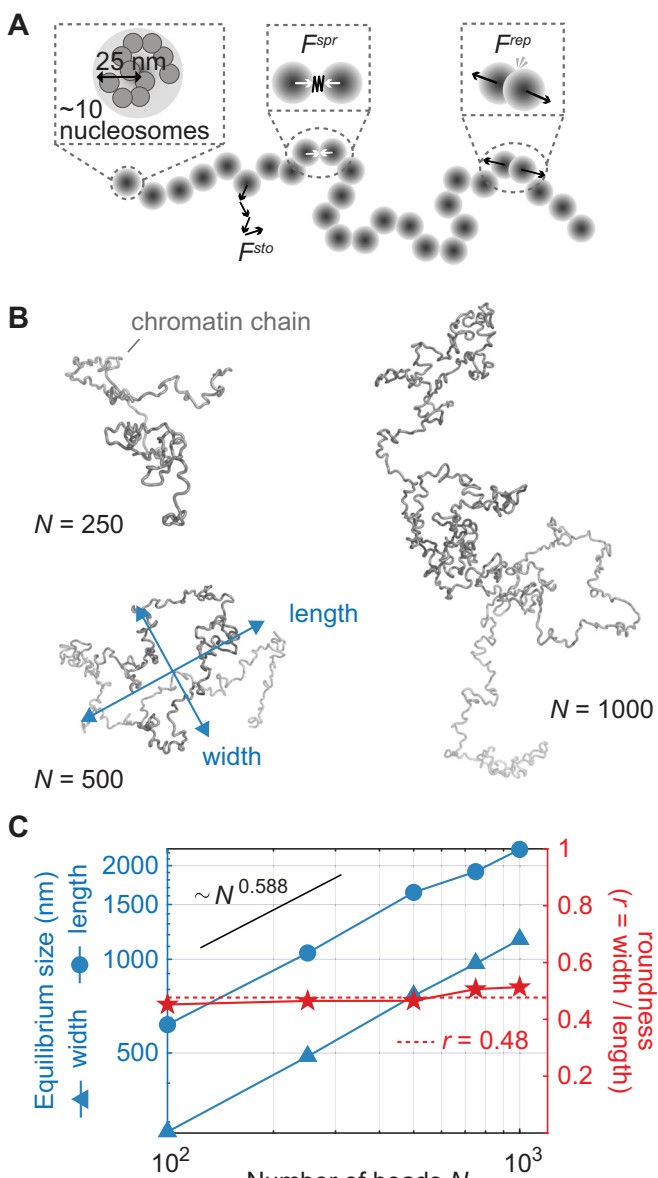

**Figure 4. A simulated chromatin chain displays theoretically expected behavior.**

(A) Schematic of a simulated chromatin chain, consisting of linked beads, each representing around 10 nucleosomes, moved by stochastic Brownian movements ($F^{sto}$), connected by a Hookean spring force ($F^{spr}$), and bead overlap cushioned by steric repulsion ($F^{rep}$). Adapted from (Gerguri et al, 2021).
(B) Snapshots of equilibrated simulated chromatin chains of varying lengths.
(C) Equilibrium length and width, as well as the width-to-length ratio, are plotted as a function of chain length. The theoretically expected scaling behavior of random self-avoiding polymers is indicated, as well as their expected roundness ($r$ = width/length).

chains, defined as their equilibrium width divided by their equilibrium length (Fig. 4C), we again see that the simulated roundness matches the theoretical prediction closely.

We conclude that, in the absence of any loop formation, our simulated chromatin chain adopts a roundness of ~0.48, as theory predicts. An unconstrained, self-avoiding polymer is therefore on

average slightly more elongated than observed chromosome arms that have reached steady state, which we have seen above have a roundness of ~0.55.

## Loop capture interactions shorten the polymer

We next investigated how loop capture interactions impact on the dimensions of our simulated chromatin chain. To model loop capture, every 10th bead is selected to be a condensin binding site, corresponding roughly to the empirically observed spacing of ~23.4 kb/11.7 beads between condensin binding sites in fission yeast. If two such condensin binding sites encounter each other by stochastic movements, a pairwise interaction forms and then persists with a defined probability, before it turns over (Fig. 5A) (Gerguri et al, 2021). Although human chromosomes are far longer, and condensin binding sites along human chromosomes are spaced much further apart (see below), here we begin by simulating the behavior of fission yeast-sized chromatin chains for computational practicality. Simulated loop capture led to the emergence of far more compact structures, seen in representative snapshots of steady state conformations found for chain lengths $N = 250$–2000 (Fig. 5B), when compared to the more extended structures of equilibrium random walks (Fig. 4B).

Note that loop interactions continue to form and break even when simulated loop capture structures have reached their final shape. Being maintained by an active, energy-consuming process, these chromatin structures can be said to have reached steady state (unlike unconstrained chromatin chains that reach an equilibrium that is maintained without energy input). Likewise, continued ATP hydrolysis cycles by condensin are required to maintain biological chromosomes (Kinoshita et al, 2015), consistent with the idea that continued capture and dissociation cycles maintain these structures.

Chromatin compaction by loop capture is mediated by the formation of characteristic rosettes that constrain the chromatin chain, as seen in previous loop capture simulations (Fig. 5B) (Cheng et al, 2015; Gerguri et al, 2021). Analyzing the average size of the resultant overall structures as a function of chain length $N$, we find a similar scaling behavior as for a self-avoiding Rouse polymer (Fig. 5C). Therefore, while more compact, loop capture results in structures that scale similarly as unconstrained random walks with respect to chain length.

When we analyze the width-to-length ratio of these more compact loop capture structures, we find that the mean roundness has increased, corresponding to a greater width-to-length ratio $r \approx 0.51$. These findings suggest that loop capture interactions reduce the polymer length more than the polymer width. The new roundness value approaches, but does not completely match, the roundness that we observed for short human chromosome arms that have reached steady state during our mitotic time course experiment ($r \approx 0.55$, Fig. 3B). These observations are consistent with the possibility that loop capture interactions contribute to shaping mitotic chromosomes, but they also suggest that additional mechanisms exist that further increase roundness, at least of short chromosome arms.

The lengths of our simulated chromatin chains are in the Mb range, corresponding in size to fission yeast chromosome arms, and we simulated a condensin binding site interval as observed in fission yeast. In comparison, human chromosome arms are an order of magnitude longer, in the tens of Mb range. Equally, condensin binding sites are found at approximately ten times greater intervals (Gerguri et al, 2021; Sutani et al, 2015). The scaling behavior of a

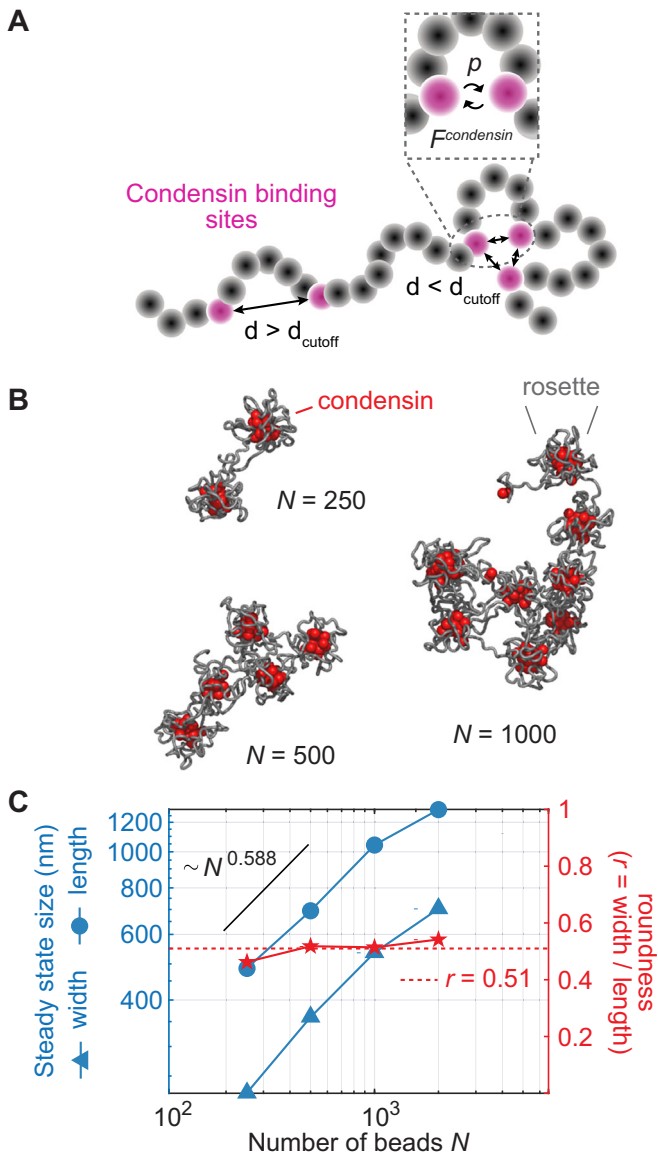

**Figure 5. Loop capture interactions compact the polymer and increase its roundness.**

(A) Schematic of loop capture interactions added to our simulated chromatin chain. If predetermined condensin binding sites come into proximity ($d < d_{cutoff}$) a bond ($F^{condensin}$) forms with probability $P$. Adapted from (Gerguri et al, 2021). (B) Snapshots of simulated chromatin chains on which condensins (red spheres) engage in loop capture interactions and which have reached steady state. (C) Steady state length and width, as well as the width-to-length ratio, are plotted as a function of simulated chain length. The theoretically expected scaling behavior of random self-avoiding polymers is indicated for comparison. The mean observed roundness ($r$ = width/length) is highlighted by a dashed red line.

Rouse polymer with excluded volume, where size grows as $N^\nu$ ($\nu = 0.588$), means this behavior is scale-free, i.e., the ratio of the size of two polymers only depends on the ratio of their lengths. Following the introduction of loop capture interactions this scaling behavior remained essentially unchanged in our simulations. We therefore postulate, but do not know for certain, that loop capture interactions affect human chromosomes of much larger dimensions in similar ways as seen for our smaller simulated chains.

## Simulated chromosome lengths and widths over time

We have so far considered simulated chromosome arms at their steady state. To conclude, we followed simulated chromosome shape changes over time. To do so, we defined an elongated initial state for our simulations, akin to that observed for natural chromosomes that appear around four times longer in prophase when compared to late mitotic stages (Fig. 2B). We therefore initialized chain conformations with a random walk four times longer than the steady state length. Upon release into simulations with loop capture interactions, these elongated chains reached a similar steady state as chains started from an unstretched, random walk conformation (for chain lengths up to $N = 2000$; Fig. EV4A). Thus, an elongated initial state does not alter the eventual steady state chromosome conformation.

We now sampled chromosome conformations over time, starting from the elongated initial state. Plotting chromosome length as a function of time recapitulated several aspects of the behavior observed of native chromosomes. Short chromatin chains quickly shortened and reached a steady state length, while longer chains took a longer time. The longest chains ($N = 4000$) had not reached steady state length at the end of our simulations, similar to what we observed for long chromosome arms during our mitotic arrest time course. In Fig. 6A, we display representative snapshots over time, for three different chain lengths. These snapshots illustrate that the shortest chain ($N = 250$) reaches a compact steady state structure within 1 min. For an intermediate size chain ($N = 1000$) it takes roughly 10 min to reach a compact steady state, while the longest chain length analyzed ($N = 4000$) remains visibly elongated and out of steady state at our last, 30 min, time point. We quantify this polymer chain behavior by plotting a time series of observed lengths over time for all chain sizes (Fig. 6B, top), which we find is well explained by an exponential relaxation process. From these fits we calculate the relaxation time (Fig. 6B, bottom), which shows an increasing relaxation time with chain length $N$. Qualitatively, the relaxation behavior resembles that found for the relaxation of native chromosomes. Short chromosome arms reach steady state quickly, while longer arms are still in the process of approaching steady state at later times. This outcome of simulated loop capture was robust to alterations in the probability of condensin loop capture (Fig. EV4B), suggesting that the timescale for diffusional relaxation of the chromatin chain dominates the observed kinetic profile.

Finally, we examined how the simulated chromosome length-to-width relationship evolves over time, plotting the measured lengths and widths of simulated chains of the five different lengths $N$. Qualitatively, the simulated loop capture chromosomes recapitulate the observed behavior of native chromosomes, where longer chromatin chains become increasingly wider over time (Fig. 6C). When we analyze the measured length-to-width relationship, similar to what we did for native chromosomes in Fig. 3C, we find that the chromosome length-to-width relationships can be described by power laws whose exponents increase over time. For simulated chromosomes, the final steady state power law exponent approaches ~0.75, a value that is the same as the exponent that we found describes shorter native chromosome arms that have reached steady state during our mitotic time course experiment.

We conclude that a simulated chromatin chain on which loop capture interactions take place undergoes shape transitions similar to those observed for native chromosomes. Shorter chains reach steady state faster than longer chains, with developing lengths and

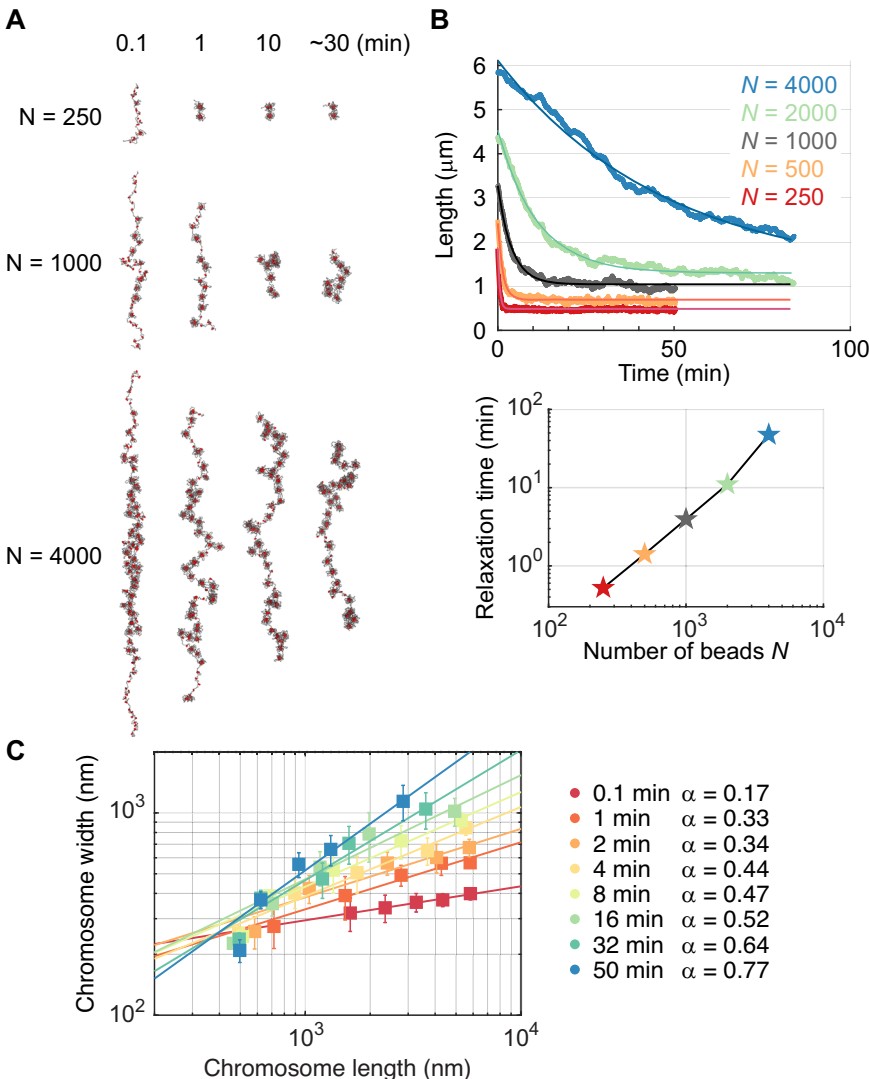

**Figure 6. Simulated loop capture interactions recapitulate a gradually increasing power law exponent of the chromosome length-to-width relationship.**

(A) Snapshots of simulated chromatin chains released from an initially elongated configuration (4× steady state length), on which condensin engages in loop capture interactions. (B) Simulated chromosome lengths were plotted over time, including exponential fits (top). The exponential fits were used to derive relaxation times, the values of which are plotted alongside as a function of chain length (bottom). (C) The chromosome length-to-width relationship over time. Each data point is the average of ten replicate simulations at the given time point, with error bars representing the standard error on the mean (errors in length are smaller than the size of symbols used). Power law fits to the length-to-width relationship are included, and the power law exponents are listed.

widths that can be described by a power law relationship with increasing exponent. These considerations put forward loop capture as a plausible mechanism that underlies, or at least contributes to, the continuous chromosome shape changes that can be observed of native chromosomes in mitotically arrested cells.

## Discussion

Our study portrays chromosomes in a fresh light. Instead of stable entities, we describe chromosomes as out-of-equilibrium structures, only the shortest of which will have typically reached their final shape by the time chromosomes split at anaphase onset. Although we acknowledge the difficulties in accurately measuring

chromosome shape, our results suggest that a *loop capture* mechanism provides a plausible explanation for the shape-shifting chromosome behavior. Longer chromosome arms become progressively wider over time. These results are in contrast to the predictions of the loop extrusion model that would result in chromosomes with a bottlebrush structure of constant width (Goloborodko et al, 2016; Samejima et al, 2025).

Chromatin loop formation by loop capture results in rosette structures, and these rosettes gradually remodel to reach a steady state that is defined by the principles of polymer physics. Longer chromatin chains occupy a volume that is not only longer, but also wider. Rosette remodeling appears to be a relatively slow process, unsurprising given the large dimensions of these objects. The process will be shaped by the time it takes for the polymer chain to

reach its steady state shape by random motion, likely with some influence by the half-life of condensin-dependent loop interactions that underpin these structures (Cheng et al, 2015; Gerlich et al, 2006; Robellet et al, 2015; Thadani et al, 2018). In addition to the rearrangement of existing rosettes, other cellular parameters might change over time as cells enter mitosis, or are blocked in an extended mitotic state. An increasing condensin concentration, histone modifications, divalent cation concentration, as well as the depletion attraction force (Hibino et al, 2024; Iida et al, 2024; Maeshima et al, 2018; Schneider et al, 2022; Vasquez et al, 2016) are all factors that contribute to chromosome formation and that could change over time. Direct condensin-condensin interactions also contribute to vertebrate chromosome formation, as do sister chromatid contacts (Chu et al, 2020; Kinoshita et al, 2022; Takahashi et al, 2016), which we omitted from our simulations. Taken together, a multitude of additional forces impact chromosome formation, and they are a possible reason for why the final observed roundness of human chromosome arms is greater than what we observe in our simple loop capture simulations. Future studies will be required to integrate these additional forces, alongside loop capture, when aiming to fully recapitulate observed chromosome behavior.

A characteristic of chromosome arms that consist of self-organizing chromatin rosettes is their gradual rounding. Rather than forming strictly cylindrical objects, rosettes arrange themselves in an elliptical volume where the middle is wider than both ends. Our simple definition of roundness, as the ratio of width divided by length, did not differentiate between ellipses and cylinders. Based on cursory visual inspection, at least shorter chromosome arms that have reached steady state are well described by ellipsoids. An additional hint that chromosome arms show ellipsoid rather than cylindrical architecture comes from chromatin interaction patterns recorded in Hi-C experiments. Chromatin contacts (other than those mediated by condensin) span greater distances in the middle of fission yeast chromosomes' arms, when compared to interactions recorded towards both ends (Kakui et al, 2017), as expected from an ellipsoid. It will be interesting to explore Hi-C datasets from metazoan mitotic chromosomes for signs of an ellipsoid shape.

While dynamic loop capture is a plausible mechanism to explain continuous shape changes, starting from an elongated appearance, an open question remains the origin of the initially elongated state. Prophase chromosomes at first are much longer than the outlines of interphase chromosome territories from which they derive (Walter et al, 2003), or than the expected equilibrium shape of an unconstrained polymer chain. The origin of the elongated initial state therefore remains to be understood. Loop extrusion by condensin could generate an elongated bottlebrush-like structure with a central condensin backbone (Goloborodko et al, 2016), however the widely scattered condensin distribution that is seen in prophase chromosomes does not lend support to this scenario (Walther et al, 2018). Short-range chromatin interactions that are established by the cohesin complex might alternatively contribute to generating an elongated shape. The reptation behavior of neighboring chromosomes within nuclear confines forms another possible contributing factor.

In higher eukaryotes, two condensin complexes, condensin I and condensin II, together shape chromosomes (Green et al, 2012; Shintomi and Hirano, 2011). Condensin I majorly affects chromosome width, while condensin II predominantly affects chromosome length. How two condensin subtypes exert apparently selective compaction in two orthogonal directions remains unknown. What is known is that condensin II engages in much farther-reaching chromatin contacts while condensin I adds shorter-range interactions (Gibcus et al, 2018). In a loop capture scenario, we can speculate that condensin II sets up a coarse rosette architecture, with condensin I inserting a layer of finer-grained rosettes afterwards (Eykelenboom et al, 2025). The larger condensin II rosettes might define the overall chromosome outline and thereby its length, while condensin I compacts surface chromatin loops that majorly affect width. Simulating chromatin behavior that arises from two distinct types of loop capture interactions provides fertile ground for future investigations.

We close by noting that, while displaying vast size differences, chromosomes from across a wide range of animals and plants all follow a universal length-to-width relationship (Kramer et al, 2021), suggesting that they are all governed by a common physical principle. The fact that chromosome arms keep shortening over time opens the possibility that "time spent in mitosis" becomes a defining factor for chromosome arm length. Future investigations of how the molecular activities of chromosomal proteins intersect with the physics-derived behavior of chromatin chains will advance our understanding of these beautiful structures.

# Methods

**Reagents and tools table**

| Reagent/resource | Reference or source | Identifier or catalog number |
| --- | --- | --- |
| **Experimental models** | | |
| HeLa (Human) | S. Narumiya, Kyoto University | NA |
| hTERT-RPE-1 (Human) | ATCC Cell Bank | Cat# CRL-4000; RRID: CVCL_4388 |
| **Chemicals, enzymes, and other reagents** | | |
| DAPI | Sigma-Aldrich | D9542 |
| RO-3306 | Enzo Life Science | ALX- 270-463-M005 |
| ProLong Gold antifade | MOP | P36930 |
| SiR-DNA | Cytoskeleton | CY-SC007 |
| Colcemid | Funakoshi | AG-CR1-3567-M005 |
| **Software** | | |
| Fiji | Open-source | http://fiji.sc |
| GraphPad Prism 10 | GraphPad | https://www.graphpad.com |
| MATLAB (vR2024a) | MathWorks | https://uk.mathworks.com |

## Cell culture, synchronization, and chromosome spreads

HeLa Kyoto cells were cultured in DMEM supplemented with 10% fetal calf serum, 0.2 mM L-glutamine, 100 U/mL penicillin and 100 μg/mL streptomycin at 37 °C in a 5% $CO_2$ environment. To study the changes in chromosome morphology, cells were synchronized at the G2/M boundary by the treatment with 9 μM

RO-3306 for 3 h, then washed once and released into medium containing 100 ng/mL colcemid. After collecting cells by mitotic shake off at the indicated times, cells were incubated with a hypotonic buffer (PBS:H$_2$O = 1:9) for 5 min, and fixed with fresh Carnoy's solution (70% methanol, 30% acetic acid). Fixed cells were dropped onto glass slides and dried. Spread chromosomes were stained with 10 µg/mL DAPI in PBS for 5 min and mounted using ProLong Gold (ThermoFisher) antifade reagent. Images were captured using a Zeiss LSM880 microscope using an ×63 objective and Airyscan detection. To image undisturbed mitotic progression, following RO-3306 release, colcemid was omitted from the release medium. DNA was visualized using the SiR DNA live stain (Cytoskeleton) and images acquired at 2.5-min intervals using a CellVoyager CQ1 High-Content Analysis System and a ×60 objective. The time from the first appearance of chromosome structure in prophase until chromosome splitting at anaphase onset was counted in 50 cells.

## Measurements of mitotic chromosome dimensions

At the 12- and 20-min time points, chromosome lengths were measured by manually drawing a line along the chromosome in Fiji, then arm widths were semiautomatically determined after selecting straight chromosome regions using a modified MATLAB script (Kakui et al, 2022). First, a mask was created by binarizing signal intensities of the original image. In this step, intensities outside the mask were set to zero. Second, a slice perpendicular to the chromosome length axis was taken from the masked image and a single Gaussian fit was applied to the slice, and the full width at half maximum (FWHM) was calculated. The previous step was repeated for all slices along the masked image, and the mean of all FWHMs was recorded as the chromosome width. In the final step, the chromosome widths were halved to obtain chromosome arm widths.

At the 30-minute and later time points, chromosome arms were traced by manual shaping in Fiji. Ellipsoid approximation was then applied in Fiji, which generates an ellipsoid of the same area as the traced shape. The lengths of the primary and secondary ellipsoid axes were recorded as chromosome arm lengths and widths. Roundness was calculated by dividing the widths by the lengths. Regression lines were plotted using the "stat_smooth" function in ggplot2. Alternatively, the same approach used for the 12- and 20-min samples was applied to the 30- to 360-min images.

## Simulations of chromatin chain behavior

Simulations were performed using a previously described chromatin simulation package (Gerguri et al, 2021), where briefly, the simulation consists of Brownian dynamics of beads connected by springs, with soft mutual repulsive interactions, such that with no further interactions the chain is a self-avoiding Rouse polymer. Each bead effectively corresponds to a radius of 25 nm, incorporating ~10 nucleosomes and roughly 2 kb of DNA. Therefore, simulations in this report, between $N = 100$ and $N = 4000$ beads, correspond to chromatin chains roughly 200 kb to 8 Mb in length. Without added loop capture interactions, these simulations are in effect "random walk" simulations, and they are initialized using a random walk configuration with random bond angles between beads on a unit sphere.

In addition, we can turn on "loop-capture" interactions during these simulations, mediated by condensin binders that are found every ten beads, roughly corresponding to the average frequency of condensin binding sites in fission yeast (Gerguri et al, 2021; Kakui et al, 2017). A small change was made to the simulation code, such as to allow random elongated ellipsoidal initial conditions, which were used for the loop capture simulations. Here, beads are placed randomly along the $z$ axis by drawing from a Gaussian distribution of standard deviation $L_0/2$, where $L_0$ is the specified initial length of the polymer. The $z$ axis positions are then sorted by position, after which each of these beads is assigned a random transverse $x$ and $y$ position, by drawing from a Gaussian distribution of standard deviation $w_0/2$, where $w_0$ is the specified initial width and depth of the polymer; since our data cannot distinguish width and depth, we let them be initially equal in our simulations. The values of $L_0$ and $w_0$ we choose for each value of $N$ the number of beads in the simulation—are roughly 4× the steady state lengths and 1× the steady state width, which we found from test simulations for each bead length. For simulations testing a different initial condition (Fig. EV4A), we used 1× the steady state lengths and 1× the steady state width, for $L_0$ and $w_0$, respectively.

The length $L$, width $w$, and depth $d$, for any conformation of the polymer in simulation is determined by calculating the eigenvalues of the covariance matrix of bead positions; if $\lambda_1, \lambda_2, \lambda_3$ are these eigenvalues ordered by decreasing magnitude $(\lambda_1 > \lambda_2 > \lambda_3)$ then $L = 2\sqrt{\lambda_1}$; $w = 2\sqrt{\lambda_2}$; $d = 2\sqrt{\lambda_3}$.

Exponential fits on simulated data were performed in MATLAB, by first using "fminsearch"—which uses the Nelder–Mead simplex method—and then using the output fit parameters as the initial condition for nonlinear regression using the "lsqcurvefit" function, which produces standard error estimates on the optimal parameters. Fits are performed on the average length and width from ten replicate simulations for each polymer length $N$ reported. The steady state length is one resultant fitting parameter, and it is this value that is reported as the average steady state length or width.

## Statistical analysis

To comprehensively measure chromosomes, the dimensions of all chromosomes or chromosome arms from two independent chromosome spreads were recorded at each time. When chromosome width was measured using Gaussian fitting, individual measurements at each pixel offset were aggregated, and the mean was recorded as the chromosome width, then halved to approximate chromosome arm width. Power law exponents were derived from the length-to-width distributions, plotted on a double logarithmic scale, by fitting linear regression lines using the "stat_smooth" function in ggplot2. Ten repeat simulations were conducted in all cases. Means and standard deviations of all measurements on simulated chains are reported. Steady state dimensions were fitted using nonlinear regression using the "lsqcurvefit" function in MATLAB.

# Data availability

The raw microscopy image data as well as numerical data used in our study are available from Figshare at https://doi.org/10.6084/m9.figshare.29380658. The MATLAB code used for semiautomatic

chromosome width measurements is available at https://github.com/FrancisCrickInstitute/Uhlman_Chromosome_Width_Finder. The code for the biophysical simulation of chromatin chain behavior was described previously (Gerguri et al, 2021) and is available from the GitHub repository (https://github.com/FrancisCrickInstitute/Chromosome-Condensation).

The source data of this paper are collected in the following database record: biostudies:S-SCDT-10_1038-S44319-025-00577-4.

## Peer review information

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

## Acknowledgements

We would like to thank Paul Bates and Eric Kramer for their input, as well as M Molodtsov and our laboratory members for discussions and critical reading of the manuscript. This work was supported by the Dr. Yoshifumi Jigami Memorial Fund, The Society of Yeast Scientists, the Institute for Fermentation, Osaka (IFO) and Waseda University grants for Special Research Projects (2021C-387, 2022C-306, 2023C-283, 2024C-285, to YKa), JSPS research grants (22K06092 to YKa, 22H04996 and 24H02283 to TH), a Wellcome Trust Investigator Award (220244/Z/20/Z to FU), and the Francis Crick Institute that receives its core funding from Cancer Research UK, the UK Medical Research Council, and the Wellcome Trust (cc2137).

## Author contributions

**Yasutaka Kakui**: Investigation; Writing—review and editing. **Yoshiharu Kusano**: Investigation; Writing—review and editing. **Tereza Clarence**: Investigation; Writing—review and editing. **Maya Lopez**: Investigation; Writing—review and editing. **Todd Fallesen**: Investigation; Writing—review and editing. **Toru Hirota**: Investigation; Writing—review and editing. **Bhavin S Khatri**: Investigation; Writing—original draft. **Frank Uhlmann**: Investigation; Writing—original draft.

Source data underlying figure panels in this paper may have individual authorship assigned. Where available, figure panel/source data authorship is listed in the following database record: biostudies:S-SCDT-10_1038-S44319-025-00577-4.

## Funding

## Disclosure and competing interests statement

The authors declare no competing interests.

# Expanded View Figures

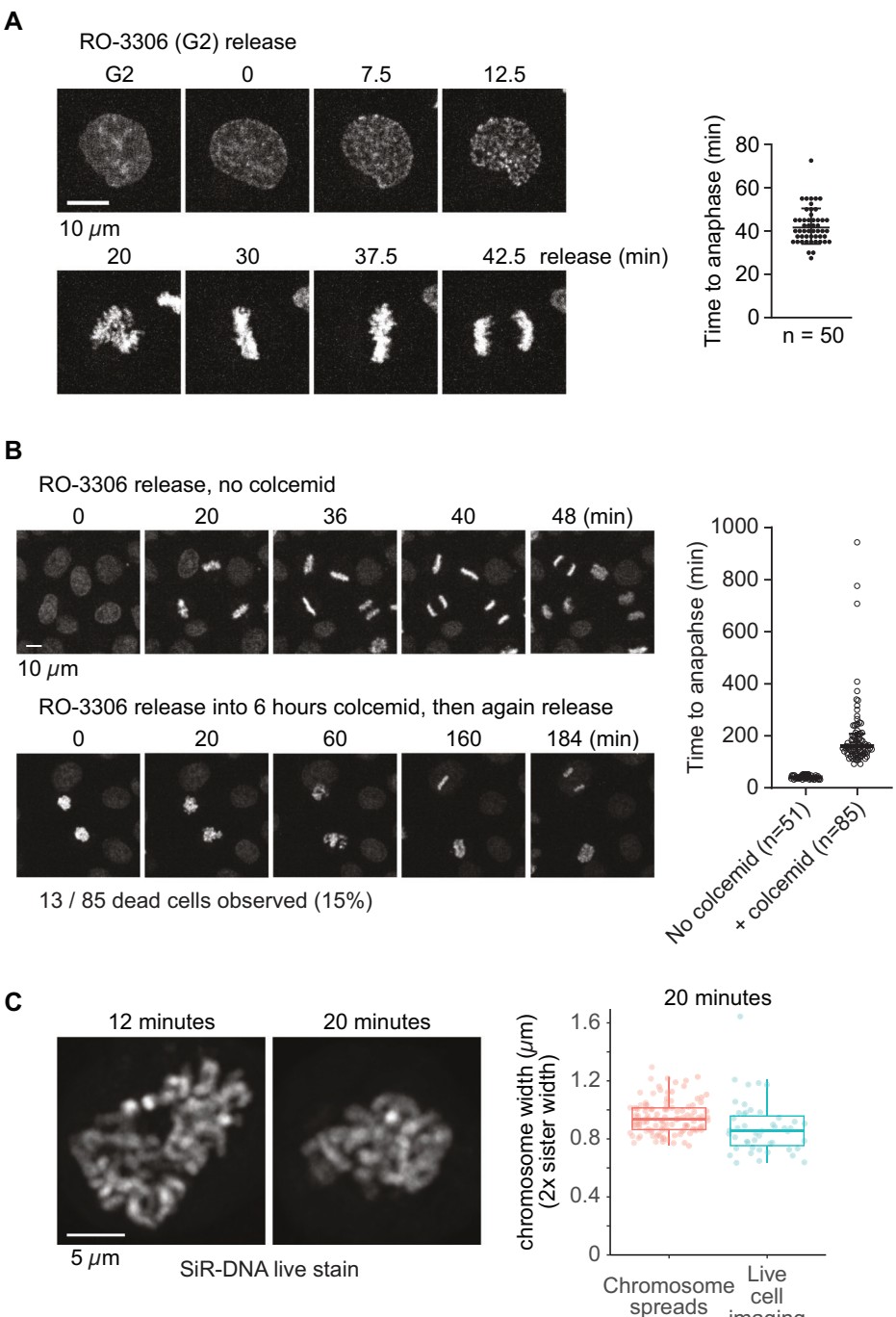

**Figure EV1.  Additional chromosome analyses during mitotic progression.**

(**A**) Timing of chromosome segregation. Cells were synchronized in G2 by RO-3306 treatment and released as in the experiment shown in Fig. 1, but in the absence of colcemid. A time series of images of a cell traversing through mitosis is shown. The time of anaphase onset was determined in 50 cells. Each measurement result is shown. The center line indicates the mean, the error bar the standard deviation. (**B**) Cells remain viable and complete cell division after release from 360 min colcemid arrest. Example time series of fields of cells are shown, either released from G2 or released following an additional 360-minute colcemid block, and the time from release to anaphase onset was recorded in the indicated numbers of cells. Each measurement result is shown. The center lines indicate the means, the error bars the standard deviations. (**C**) Chromosomes visualized in live cells using the SiR-DNA stain, following release from G2 synchronization by RO-3306. Example images are shown at 12 and 20 min after release. Chromosome widths measured at 20 min after release are compared to those measured on fixed and spread DAPI-stained chromosomes (compare Figs. 1 and 2B). The box plots show the medians (center), interquartile ranges (bounds of boxes) and 90th percentile ranges (whiskers). The slightly wider appearance of chromosomes after spreading could arise not from distortion, but if chromosomes whose diameter is not perfectly round oriented themselves with their flat side on the surface.

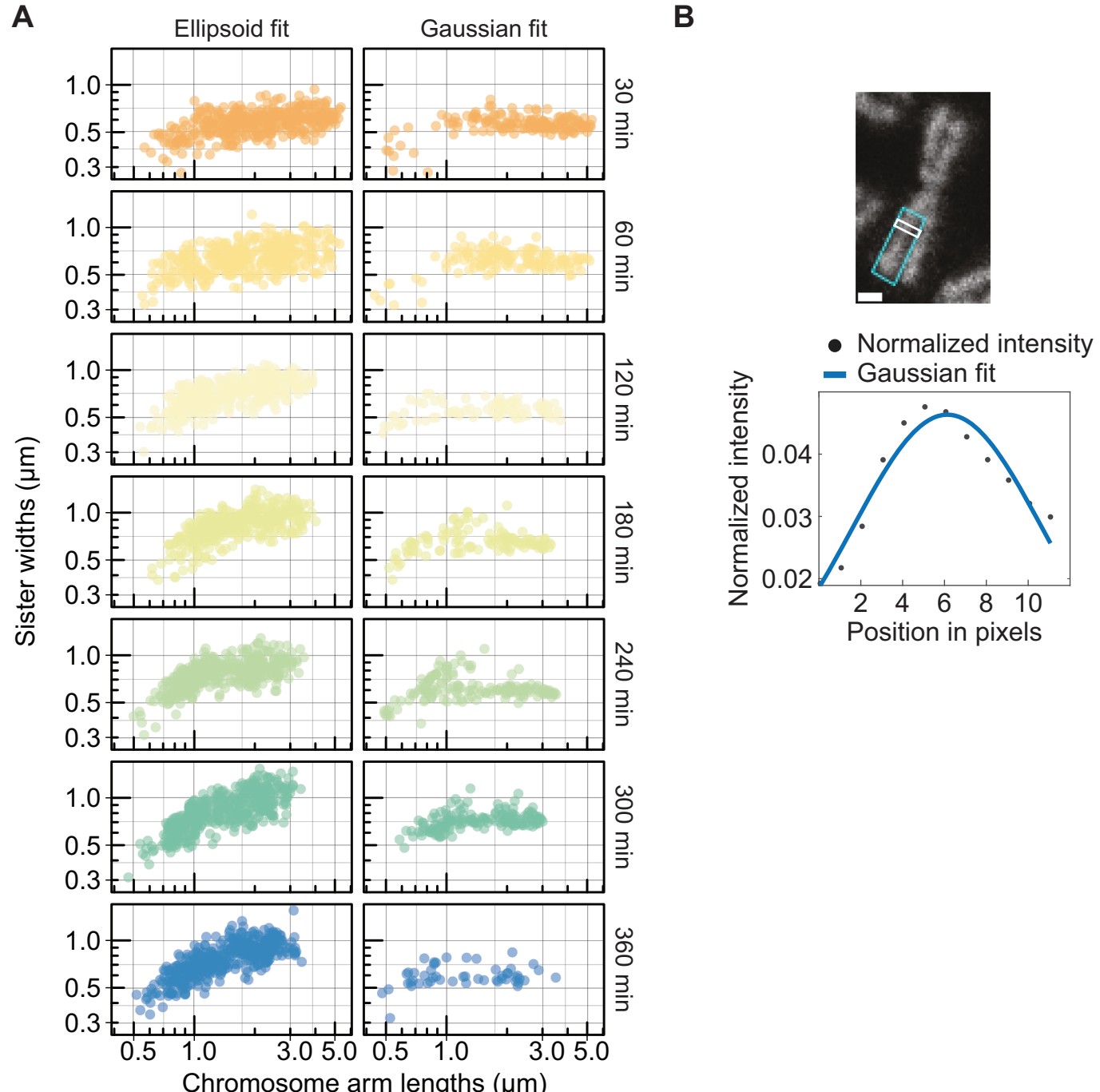

**Figure EV2. Comparison of chromosome width measurements by ellipsoid and moving Gaussian fitting.**

(A) Ellipsoid fit measurements reported in Fig. 2 are compared to measurements obtained by a sliding Gaussian fit of a subset of the same chromosome arms. Both methods reveal gradual chromosome shortening and widening over time, and that longer arms become progressively wider than shorter arms. The Gaussian fits suggest that the width plateaus with respect to length, for the longest arms. A plateau width of the longest arms is compatible with the idea that these arms widen at a uniform rate and are still on course to approach, but have not yet reached, their final steady state width. However, the incomplete sampling makes it difficult to quantitively compare the two sets of measurements, thereby preventing a firm conclusion. (B) A limitation of the sliding Gaussian approach is the fact that DAPI intensity from the neighboring sister chromatid often distorts the fit (see the example—scale bar, 1 μm). For this reason, and because other chromosome arms had to be entirely excluded from the sliding Gaussian fit approach if they were bent, we use the ellipsoid fit measurements that could be applied to all chromosome arms for subsequent analyses.

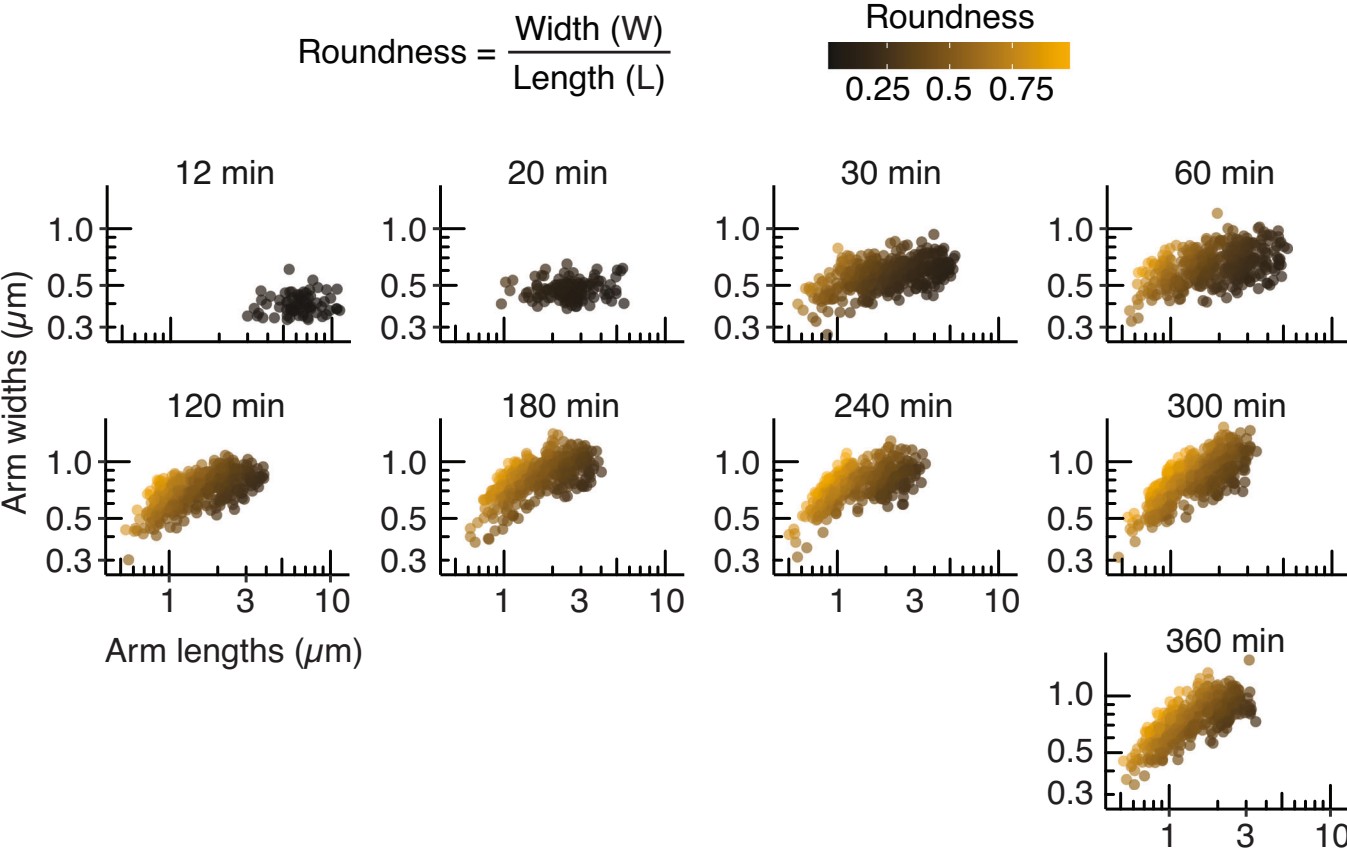

**Figure EV3. Chromosome roundness over time.**

Chromosome width as a function of length, at the designated times are shown, with increasing roundness indicated as hues of black to yellow.

**A**

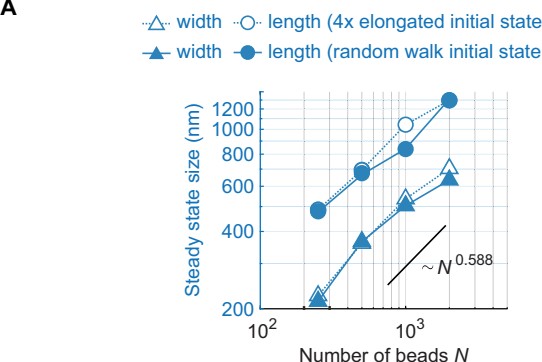

**B**

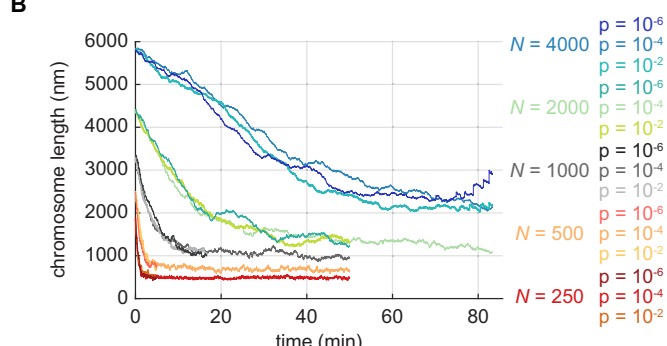

**Figure EV4. Comparison of different initial states, and looping probabilities, on the chromatin simulations.**

(A) Steady state chromosome lengths and widths, when simulations of increasing chain lengths were started from either an unstretched random conformation, or from a four times elongated conformation. The theoretically expected scaling behavior of random self-avoiding polymers is indicated for comparison. (B) The time change of simulated polymer dimensions is robust to changes in condensin's loop capture probability. As Fig. 6A, but loop capture probabilities two magnitudes higher or lower than the standard probability were simulated.

