## [Peer Review File · EMBO Reports]

Progressive chromosome shape changes during cell divisions

Yasutaka Kakui, Yoshiharu Kusano, Tereza Clarence, Maya Lopez, Todd Fallesen, Toru Hirota, Bhavin Khatri, and Frank Uhlmann

Corresponding author(s): Frank Uhlmann (frank.uhlmann@crick.ac.uk) , Bhavin Khatri (b.khatri@imperial.ac.uk)

Review Timeline:

Transfer Date:	9th Mar 25
Editorial Decision:	19th Mar 25
Revision Received:	29th Jun 25
Editorial Decision:	23rd Jul 25
Revision Received:	29th Jul 25
Accepted:	21st Aug 25

Editor: Deniz Senyilmaz Tiebe

Transaction Report: This manuscript was transferred to EMBO reports following peer review at Review Commons.

Review
COMMONS

Review #1

1. Evidence, reproducibility and clarity:

Evidence, reproducibility and clarity (Required)

****Summary:**** This manuscript authored by Kakui and colleagues aims to understand on how mitotic chromosomes get their characteristic, condensed X shape, which is functionally important to ensure faithful chromosome segregation and genome inheritance to both daughter cells. The authors focus on the condensin complex, a central player in chromosome condensation. They ask whether it condenses chromosomes through a now broadly popular "loop-extrusion" mechanism, in which a chromatin-bound condensin complex reels chromatin into loops until it dissociates or encounters a roadblock on the polymer (another condensin or some other protein complex), or through an alternative, "diffusion-capture" mechanism, in which a chromatin-bound condensin complex forms loops by encountering another chromatin-bound condensin until they dissociate from DNA (or from each other.)

The authors measured the progressive changes in the shape of mitotic chromosomes by taking samples at given time points from synchronized and mitotically arrested cells and found that while all chromosomes become more condensed and shorter, their width correlated with the length of the chromosome arms. They also observed that chromosome compaction/shortening evolves on a time scale much longer than the interval between the onset of chromosome condensation and the start of chromosome segregation, suggesting that chromatin condensation does not reach its steady-state during an unperturbed mitosis. The observed width-length correlation could be described by a power law with an exponent that increases with the time (i.e. chromosome condensation). The authors also performed polymer simulations of the diffusion-capture mechanism and found that the simulations semi-quantitatively recapitulate their experimental observations.

****Major Comments****

My most substantial comments focus on somewhat technical details of the image analysis approaches taken and the polymer models employed. However, as all reported data are derived from those details, I feel it is crucial to address them.

1. Definition/measurement of chromatin arms width and length. The approach taken to manually threshold an "arm" object and then fitting it with a same-area ellipse is not an ideal approach to gauge length and width of the arm, for the following reasons: (1) An ellipse appears to do a poor job approximating many of the objects that we see in the

zoom-in insets of Fig. 1. Importantly, for somewhat bent shapes we see in the insets it likely strongly underestimates the length of the arms; this approach also presents potential problems for measuring width as well (see 2 and 3 here). (2) One concern is that, due to the diffraction limit, a cylindrical fluorescent object could appear somewhat wider at the mid-length than the real underlying cylinder or the poles; this effect could become more pronounced as the object gets brighter and shorter. (3) Forcing the fit to an ellipse to objects that are not truly rod-shaped can drive an overestimation of the width of the object, and I suspect that this effect also might correlate with the length and brightness of the object. (4) Given 1-3 above, I think the approach the authors used for the first two time points, while not perfect, is better suited and likely more robust while avoiding these caveats. Moreover, why the authors cannot use this same approach (but just for each arm separately) for the later (30+ min) time points as they used for first two is unclear. This point is underscored by the observation that there is a drastic difference in the results between the first two and all subsequent points. When the authors compared the two approaches at the 30 min time point (where width-length dependence is still weak) in different cell lines they did indeed see different results (Fig. S2), although they concluded that the difference was acceptable.

Along these lines, the difference between short and long arms for the chromosome in the insets of Fig. 1 are quite subtle, except maybe at 180 and 240 min. On a related note, it might be informative to compare data for the two sister chromatid arms (as the underlying polymer has the same length) long vs long and short vs short and long vs short to help establish the robustness of the approach.

2. Regarding the power-law distribution, it is hard to judge based on the presented data whether it is a really good description of the data or not. In Fig. 1c, the points for a given time can barely be distinguished, while in Fig. 1b the authors plot individual time points in the panels, but the fits and points are overlapping so much that it is challenging to the main trends described by the clouds. The most informative approach for the reader would be to provide confidence intervals of the best fit parameters for all parameters that were varied in the fit. As the authors make some conclusions based on the power-law exponent values they observed, it would be helpful to know how confident we are in those values.

3. The conclusion that short arms equilibrate faster based on Fig. 3a is not fully convincing. For example, in a scenario where ~ 1.5 microns is the equilibrium length for all arms, and that the longest arms equilibrate the fastest - you would see the same qualitative pattern for quantiles, not much change in low percentiles, while you would observe a decrease in the values for the high percentiles. The authors might be right, but Fig. 3A does not unambiguously demonstrate that it is so based on this evidence alone.

4. As for chromosome roundness, typically in image analysis, roundness is defined through the ratio of $(\text{perimeter})^2/\text{area}$; it might be better to use "aspect ratio" for the metrics used by

the authors. And, perhaps, one should expect that shorter (measured, not necessarily by polymer contour length) arms should have a higher width/length ratio? If one selects for more round objects, there should be no surprise that the width and length get almost proportional. Given all of this, I am not sure whether width/aspect ratio serves as a good proxy for the chromatin condensation progression, which is how the authors are employing this data in the manuscript as written.

5. For the diffusion-capture model simulations, I think the results of the simulation would strongly depend on the assumptions of the probability to associate and the time scale of dissociation of the beads representing the condensin complex. For example, for a very strong association one might expect that all condensin will end up in one big condensate, even in the case of a long polymer. This is not explored/discussed at all. Did the authors optimize their model in any way? If not, how have they estimated the values they used? Moreover, perhaps this is an opportunity to learn/predict something about condensin properties, but the authors do not take advantage of this opportunity.

In addition, the authors did some checks to show that the steady-state results of the simulations do not depend on the initial conditions. However, as some of the results reported concern the polymer evolution to the steady state (Fig.6b-c), they also need to examine whether these results depend on the chosen initial conditions (or not), and if they do, what is the rationale for the choices the authors have made?

6. A more thorough discussion of other possible models, beyond diffusion-capture model considered here, would be beneficial to the reader. First, the authors practically discard the possibility of the loop-extrusion model to explain their observations (although they never explicitly state this in the abstract or discussion). However, they neither leveraged simulations to rigorously compare models nor included some other substantiated arguments to explain why they prefer their model. This is important, as one of the major findings here is that the chromatin never reaches steady state for condensation, making it challenging to intuit what one should expect in this very dynamic state. Second, the authors, while briefly mentioning that there might be some other mechanisms contributing to the mitotic chromosome reshaping, do not really discuss those possibilities in a scholarly way. For example, work by the Kleckner group has suggested an involvement of bridges between sister chromatids into their shortening dynamics (Chu et al. Mol Cell 2020). Third, the authors do not discuss how they envision the interplay between the different SMC complexes - cohesin, condensin I and condensin II - as they act on the same chromatin polymer, or at least acknowledge a possible role that this interplay might contribute to the observed time dependencies.

2. Significance:

Significance (Required)

The question the authors are trying to address is fundamental and important. While loop extrusion-driven mitotic chromosome organization is a popular model, considering alternative models is always crucial, especially when one can find experimental observations that allow us to discriminate between possible models. The main limitations are: 1) the performance of the approach the authors take to measure chromosome shape is in question and 2) the main competitive model (loop extrusion) is not modeled. If all shortcomings are addressed this work may provide strong evidence for the diffusion-capture model and thus advance our mechanistic understanding of mitotic processes, which will be of broad interest to the fields of genome and chromosome biology.

3. How much time do you estimate the authors will need to complete the suggested revisions:

Estimated time to Complete Revisions (Required)

(Decision Recommendation)

Between 3 and 6 months

4. Review Commons values the work of reviewers and encourages them to get credit for their work. Select 'Yes' below to register your reviewing activity at Web of Science Reviewer Recognition Service (formerly Publons); note that the content of your review will not be visible on Web of Science.

No

Review #2

1. Evidence, reproducibility and clarity:

Evidence, reproducibility and clarity (Required)

****Summary****

The authors tracked the progression of mitotic chromosome compaction over time by imaging chromatin spreads from HeLa cells that were released from G2/M arrest. By measuring the mitotic chromosome arms' width and length at different times post-release, the authors demonstrated that the speed at which the chromosome arms reach an

equilibrium state is dependent on their length. The authors were able to recapitulate this observation using polymer simulations that they previously developed, supporting the model of loop capture as the mechanism for mitotic chromosome compaction.

****Main Comments****

This is a straightforward paper that supports an alternative mechanism (relative to the highly popular loop-extrusion) model for chromosome compaction. My comments are meant to help the manuscript reach a wider audience.

I suggest that "equilibrium" be replaced with "equilibrium length" since it is the only equilibrium parameter of concern.

In the results, it may help to describe how loop capture and loop extrusion are incorporated into the simulations, using terminology that non-experts can understand. Such a description should be accompanied by figures that can be related to the other figures (color scheme, nomenclature if possible).

****Other comments****

P5: Is it possible the chromosome-spread processing may distort the structures of the chromosomes?

Please clarify whether mitosis can complete after drug removal at the various treatment intervals.

P6: "Our records are not, therefore, meant as an accurate absolute measure of individual arms. Rather, fitting allows us to sample all chromosome arms and deduce overall trends of chromosome shape changes over time"

It would be better to state this sentence earlier in this paragraph, or earlier in the section so that readers' expectations are curbed when they're reading the detailed analysis plan.

P6: "As soon as individual chromosome arms become discernible (30 minutes), longer chromosome arms were wider, a trend that became more pronounced as time progressed." Implies that at early time points, when the lengths of the arms were unknown, the longer arms were equal or narrower than the short arms. I think it's more accurate to say that as soon as the arms were resolved, the longer arms appeared wider.

P7: Is there a functional consequence to the long arms not equilibrating before anaphase onset?

P13: "In a loop capture scenario, we can envision how condensin II sets up a coarse rosette architecture, with condensin I inserting a layer of finer-grained rosettes."

This should be illustrated in a figure.

****Figures****

Fig. 1: "...while insets show chromosomes at increasing magnification over time" sounds like the microscope magnification is changing over time. Please change "magnification" to "enlargement". Alternatively, if the goal of the figure is to illustrate the shape/dimensions change of the chromosomes over time, wouldn't it be better to keep all the enlargements at the same scale?

Fig. 2a plot: Does the distribution of normalized intensities really justify a Gaussian fit? I see a double Gaussian.

Please label the structures that resemble "rosettes".

Lu Gan

2. Significance:

Significance (Required)

General This is a simulation-centric study of mammalian chromosome compaction that supports the loop-capture mechanism. It may be viewed as provocative by some readers because loop-extrusion has dominated the chromosome-compaction literature in the past decade. The only limitation, which is best addressed by future studies, is the absence of more direct molecular evidence of loop capture in situ. Though this same limitation applies to studies of the loop-extrusion mechanism.

Advance It is valuable for the field to consider alternative mechanisms. In my opinion, the dominant one has been studied to death by indirect methods without a direct molecular-resolution readout in situ. While the field awaits better experimental tools, more mechanisms should be explored.

Audience The chromosome-biology community (both bacterial and eukaryotic) will be interested.

Expertise My lab uses cryo-ET to study chromatin in situ.

3. How much time do you estimate the authors will need to complete the suggested revisions:

Estimated time to Complete Revisions (Required)

(Decision Recommendation)

Less than 1 month

4. Review Commons values the work of reviewers and encourages them to get credit for their work. Select 'Yes' below to register your reviewing activity at Web of Science Reviewer Recognition Service (formerly Publons); note that the content of your review will not be visible on Web of Science.

No

Review #3

1. Evidence, reproducibility and clarity:

Evidence, reproducibility and clarity (Required)

In this manuscript, Kakui et al. measured the length/width relationships of mitotic chromosomes in human cells that had entered mitosis for different durations. This simple measurement revealed very interesting behaviors of mitotic chromosomes. They found that the longer chromosome arms were wider than shorter ones. Mitotic chromosomes became progressively wider over time, with shorter ones reached the final state faster than the longer ones. They then built a loop-capture polymer model, which explained the time-dependent increase of width/length ratio rather well, but did not quite explain the final roundness of chromosomes.

I suggest the following points for the authors to consider.

****Major points****

1. There is no experimental evidence that the loop capture mechanism is condensin-dependent. Can the authors deplete condensin I or II or both and measure chromosome length and width in similar assays? This will link their models to molecular players.
2. It seems rather intuitive to me that if one defines the spacing of the condensin-binding sites, then the loop sizes will be the same between shorter and longer chromosomes. It then follows that shorter chromosomes are rounder. Is it that simple? If not, can the authors provide a better explanation.
3. If the loop sizes are the same between shorter and longer chromosomes, why can't the loop extrusion model explain this phenomenon? If one assumes that condensin is stopped by the same barrier element and has the same distribution at the loop base, this should produce the same outcome as loop capture.

****Minor points****

1. "We are aware that this approximation underestimates the length of the longest chromosome arms and overestimates the length of the shortest arms." should be "We are aware that this approximation underestimates the length of the longer chromosome (q) arms and overestimates the length of the shorter (p) arms.". Right?
2. Some scientists argue that the final chromosome conformation might be kinetically driven. Even if the short chromosomes have reached the final roundness, this doesn't necessarily mean that they have reached equilibrium in cells. "Steady state" might be a better term to describe the chromosomes in vivo, as there are clearly energy-consuming processes.

2. Significance:

Significance (Required)

I find the paper intellectually stimulating and a pleasure to read. It suggests a plausible explanation for mitotic chromosome formation. As such, it will be of great interest to scientists in the chromatin field.

3. How much time do you estimate the authors will need to complete the suggested revisions:

Estimated time to Complete Revisions (Required)

(Decision Recommendation)

Between 3 and 6 months

4. Review Commons values the work of reviewers and encourages them to get credit for their work. Select 'Yes' below to register your reviewing activity at Web of Science Reviewer Recognition Service (formerly Publons); note that the content of your review will not be visible on Web of Science.

Yes

Review #4

1. Evidence, reproducibility and clarity:

Evidence, reproducibility and clarity (Required)

The take home message of this study is that chromosome structure can be attained through mechanisms of looping that do not require an explicit loop extrusion function. As the authors states, alternative models of loop capture have been proposed, dating from 2015-2016. These models show DNA chains through simply Brownian diffusion can adopt a loop structure (citation 27, 28 and similarly Entropy gives rise to topologically associating domains Vasquez et al 2016 DOI: 10.1093/nar/gkw510).

In this study, the authors go through careful and well-documented chromosome length measurements through prophase and metaphase. The modeling studies clearly show that loop capture provides a tenable mechanism that accounts for the biological results. The results are clearly written and propose an important alternative narrative for the foundation of chromosome organization.

2. Significance:

Significance (Required)

The study is important because it takes a reductionist approach using just Brownian motion and loop capture to ask how well the fundamental processes will recapitulate the biological outcome. The fact that loop capture can account for the arm length to width relationships on biological time scales is important to report to the community.

The work is extremely well done and the analysis of chromosome features is thorough and well-documented.

3. How much time do you estimate the authors will need to complete the suggested revisions:

Estimated time to Complete Revisions (Required)

(Decision Recommendation)

Less than 1 month

Yes

Revision Plan

Manuscript number: RC -2025-02864

Corresponding author(s): Frank Uhlmann

1. General Statements

We thank the four reviewers for their interest in our study and for their constructive comments.

2. Description of the planned revisions

Reviewer #1 (Evidence, reproducibility and clarity (Required)):

Summary: This manuscript authored by Kakui and colleagues aims to understand on how mitotic chromosomes get their characteristic, condensed X shape, which is functionally important to ensure faithful chromosome segregation and genome inheritance to both daughter cells. The authors focus on the condensin complex, a central player in chromosome condensation. They ask whether it condenses chromosomes through a now broadly popular "loop-extrusion" mechanism, in which a chromatin-bound condensin complex reels chromatin into loops until it dissociates or encounters a roadblock on the polymer (another condensin or some other protein complex), or through an alternative, "diffusion-capture" mechanism, in which a chromatin-bound condensin complex forms loops by encountering another chromatin-bound condensin until they dissociate from DNA (or from each other.)

The authors measured the progressive changes in the shape of mitotic chromosomes by taking samples at given time points from synchronized and mitotically arrested cells and found that while all chromosomes become more condensed and shorter, their width correlated with the length of the chromosome arms. They also observed that chromosome compaction/shortening evolves on a time scale much longer than the interval between the onset of chromosome condensation and the start of chromosome segregation, suggesting that chromatin condensation does not reach its steady-state during an unperturbed mitosis. The observed width-length correlation could be described by a power law with an exponent that increases with the time (i.e. chromosome condensation). The authors also performed polymer simulations of the diffusion-capture mechanism and found that the simulations semi-quantitatively recapitulate their experimental observations.

Major Comments

My most substantial comments focus on somewhat technical details of the image analysis approaches taken and the polymer models employed. However, as all reported data are derived from those details, I feel it is crucial to address them.

We thank the reviewer for their suggestions on how to improve our image analysis and polymer modelling experiments. We are keen to develop both aspects of our manuscript with additional experiments as detailed below.

1. Definition/measurement of chromatin arms width and length. The approach taken to manually

Revision Plan

threshold an "arm" object and then fitting it with a same-area ellipse is not an ideal approach to gauge length and width of the arm, for the following reasons: (1) An ellipse appears to do a poor job approximating many of the objects that we see in the zoom-in insets of Fig. 1. Importantly, for somewhat bent shapes we see in the insets it likely strongly underestimates the length of the arms; this approach also presents potential problems for measuring width as well (see 2 and 3 here). (2) One concern is that, due to the diffraction limit, a cylindrical fluorescent object could appear somewhat wider at the mid-length than the real underlying cylinder or the poles; this effect could become more pronounced as the object gets brighter and shorter. (3) Forcing the fit to an ellipse to objects that are not truly rod-shaped can drive an overestimation of the width of the object, and I suspect that this effect also might correlate with the length and brightness of the object. (4) Given 1-3 above, I think the approach the authors used for the first two time points, while not perfect, is better suited and likely more robust while avoiding these caveats. Moreover, why the authors cannot use this same approach (but just for each arm separately) for the later (30+ min) time points as they used for first two is unclear. This point is underscored by the observation that there is a drastic difference in the results between the first two and all subsequent points. When the authors compared the two approaches at the 30 min time point (where width-length dependence is still weak) in different cell lines they did indeed see different results (Fig. S2), although they concluded that the difference was acceptable.

While the manuscript was under review, we have developed an improved pipeline to measure chromosome widths. As suggested by the reviewer, this approach is based on the method used for the first two time points. An additional improvement allows us to take automated measurements along the entire chromosome arm length, instead of being restricted to straight segments. We propose to use the improved algorithm to repeat all measurements at later time points.

Along these lines, the difference between short and long arms for the chromosome in the insets of Fig. 1 are quite subtle, except maybe at 180 and 240 min. On a related note, it might be informative to compare data for the two sister chromatid arms (as the underlying polymer has the same length) long vs long and short vs short and long vs short to help establish the robustness of the approach.

The chromosome arm width differences are clear and measurable. We will select insets that illustrate the arm width differences in a more representative way, and we will furthermore conduct the suggested analyses on subsets of chromosome arms to test the robustness of our approach.

2. Regarding the power-law distribution, it is hard to judge based on the presented data whether it is a really good description of the data or not. In Fig. 1c, the points for a given time can barely be distinguished, while in Fig. 1b the authors plot individual time points in the panels, but the fits and points are overlapping so much that it is challenging to the main trends described by the clouds. The most informative approach for the reader would be to provide confidence intervals of the best fit parameters for all parameters that were varied in the fit. As the authors make some conclusions based on the power-law exponent values they observed, it would be helpful to know how confident we are in those values.

Revision Plan

Confidence intervals of the power law exponents will be provided.

3. The conclusion that short arms equilibrate faster based on Fig.3a is not fully convincing. For example, in a scenario where ~1.5 microns is the equilibrium length for all arms, and that the longest arms equilibrate the fastest - you would see the same qualitative pattern for quantiles, not much change in low percentiles, while you would observe a decrease in the values for the high percentiles. The authors might be right, but Fig. 3A does not unambiguously demonstrate that it is so based on this evidence alone.

Our reasoning is based on the observation that the shortest percentiles do not change or do not change rapidly after 30 minutes, while the longest percentiles are clearly still relaxing towards a steady state. We will repeat this analysis with the new measurements, obtained in response to point 1.

4. As for chromosome roundness, typically in image analysis, roundness is defined through the ratio of (perimeter)²/area; it might be better to use "aspect ratio" for the metrics used by the authors. And, perhaps, one should expect that shorter (measured, not necessarily by polymer contour length) arms should have a higher width/length ratio? If one selects for more round objects, there should be no surprise that the width and length get almost proportional. Given all of this, I am not sure whether width/aspect ratio serves as a good proxy for the chromatin condensation progression, which is how the authors are employing this data in the manuscript as written.

We thank the reviewer for alerting us to an alternatively used definition of 'roundness'. We will consider this concern, with one solution being to use 'width-length ratio' in its place.

5. For the diffusion-capture model simulations, I think the results of the simulation would strongly depend on the assumptions of the probability to associate and the time scale of dissociation of the beads representing the condensin complex. For example, for a very strong association one might expect that all condensin will end up in one big condensate, even in the case of a long polymer. This is not explored/discussed at all. Did the authors optimize their model in any way? If not, how have they estimated the values they used? Moreover, perhaps this is an opportunity to learn/predict something about condensin properties, but the authors do not take advantage of this opportunity.

We in fact explored the consequences of altering diffusion capture on and off rates when we initially developed the loop capture simulations, and we will report on the robustness of our model to the probability of dissociation as part of our revisions.

In addition, the authors did some checks to show that the steady-state results of the simulations do not depend on the initial conditions. However, as some of the results reported concern the polymer evolution to the steady state (Fig.6b-c), they also need to examine whether these results depend on the chosen initial conditions (or not), and if they do, what is the rationale for the choices the authors have made?

Revision Plan

The current manuscript contains a comparison of steady states reached after simulations were started from elongated or random walk initial states (see Supplementary Figure 4). We will provide better justification for the choice of a 4x elongated initial state, which approximates the initial state observed *in vivo*.

6. A more thorough discussion of other possible models, beyond diffusion-capture model considered here, would be beneficial to the reader. First, the authors practically discard the possibility of the loop-extrusion model to explain their observations (although they never explicitly state this in the abstract or discussion). However, they neither leveraged simulations to rigorously compare models nor included some other substantiated arguments to explain why they prefer their model. This is important, as one of the major findings here is that the chromatin never reaches steady state for condensation, making it challenging to intuit what one should expect in this very dynamic state. Second, the authors, while briefly mentioning that there might be some other mechanisms contributing to the mitotic chromosome reshaping, do not really discuss those possibilities in a scholarly way. For example, work by the Kleckner group has suggested an involvement of bridges between sister chromatids into their shortening dynamics (Chu et al. Mol Cell 2020). Third, the authors do not discuss how they envision the interplay between the different SMC complexes - cohesin, condensin I and condensin II - as they act on the same chromatin polymer, or at least acknowledge a possible role that this interplay might contribute to the observed time dependencies.

The reviewer raises important points, which we are keen to explore by performing loop extrusion simulations, as well as in an expanded discussion section.

Reviewer #1 (Significance (Required)):

Significance:

The question the authors are trying to address is fundamental and important. While loop extrusion-driven mitotic chromosome organization is a popular model, considering alternative models is always crucial, especially when one can find experimental observations that allow us to discriminate between possible models. The main limitations are: 1) the performance of the approach the authors take to measure chromosome shape is in question and 2) the main competitive model (loop extrusion) is not modeled. If all shortcomings are addressed this work may provide strong evidence for the diffusion-capture model and thus advance our mechanistic understanding of mitotic processes, which will be of broad interest to the fields of genome and chromosome biology.

We are happy to hear that the reviewer agrees that our work ‘*may provide strong evidence for the diffusion-capture model and thus advance our mechanistic understanding of mitotic processes*’. See above for how we propose to address the two main limitations.

Revision Plan

Reviewer #2 (Evidence, reproducibility and clarity (Required)):

SUMMARY

The authors tracked the progression of mitotic chromosome compaction over time by imaging chromatin spreads from HeLa cells that were released from G2/M arrest. By measuring the mitotic chromosome arms' width and length at different times post-release, the authors demonstrated that the speed at which the chromosome arms reach an equilibrium state is dependent on their length. The authors were able to recapitulate this observation using polymer simulations that they previously developed, supporting the model of loop capture as the mechanism for mitotic chromosome compaction.

MAIN COMMENTS

This is a straightforward paper that supports an alternative mechanism (relative to the highly popular loop-extrusion) model for chromosome compaction. My comments are meant to help the manuscript reach a wider audience.

I suggest that "equilibrium" be replaced with "equilibrium length" since it is the only equilibrium parameter of concern.

The reviewer is correct, and we will implement this change, also taking into account the reasoning of reviewer 3 that 'steady state' is a better term to describe a final shape that is maintained by an active process.

In the results, it may help to describe how loop capture and loop extrusion are incorporated into the simulations, using terminology that non-experts can understand. Such a description should be accompanied by figures that can be related to the other figures (color scheme, nomenclature if possible).

Following from the reviewer's suggestion, we will provide schematics of the loop capture and loop extrusion mechanisms.

OTHER COMMENTS

P5: Is it possible the chromosome-spread processing may distort the structures of the chromosomes?

We will compare chromosome dimension in live cells with those following spreading to investigate this possibility.

Please clarify whether mitosis can complete after drug removal at the various treatment intervals.

Drug treatment and removal is often used as an experimental tool. We will perform a control experiment to explore whether mitosis can indeed complete after drug removal under our experimental conditions.

P6: "Our records are not, therefore, meant as an accurate absolute measure of individual arms. Rather, fitting allows us to sample all chromosome arms and deduce overall trends of

Revision Plan

chromosome shape changes over time"

It would be better to state this sentence earlier in this paragraph, or earlier in the section so that readers' expectations are curbed when they're reading the detailed analysis plan.

Note that we will employ an additional image analysis method, in response to comments from reviewer 1, which should lead to more reliable width measurements.

P6: "As soon as individual chromosome arms become discernible (30 minutes), longer chromosome arms were wider, a trend that became more pronounced as time progressed." Implies that at early time points, when the lengths of the arms were unknown, the longer arms were equal or narrower than the short arms. I think it's more accurate to say that as soon as the arms were resolved, the longer arms appeared wider.

We will adopt the reviewers' more accurate wording.

P7: Is there a functional consequence to the long arms not equilibrating before anaphase onset?

The reviewer raises an interesting question, which we will explore in our revised discussion. One consequence of not reaching 'steady state' is that 'time in mitosis' becomes a key parameter that defines compaction at anaphase onset.

P13: "In a loop capture scenario, we can envision how condensin II sets up a coarse rosette architecture, with condensin I inserting a layer of finer-grained rosettes."

This should be illustrated in a figure.

We will consider such a figure, though the roles of two condensin complexes is peripheral to our current study. Investigating the consequences of two distinct condensins for chromosome formation will provide fertile ground for future investigations.

FIGURES

Fig. 1: "...while insets show chromosomes at increasing magnification over time" sounds like the microscope magnification is changing over time. Please change "magnification" to "enlargement". Alternatively, if the goal of the figure is to illustrate the shape/dimensions change of the chromosomes over time, wouldn't it be better to keep all the enlargements at the same scale?

During the revisions, we will explore whether to show the insets at the same magnification, or to adjust the wording as suggested by the reviewer.

Fig. 2a plot: Does the distribution of normalized intensities really justify a Gaussian fit? I see a double Gaussian.

The chosen example indeed resembles a double Gaussian. We will explore whether this is due to noise in the measurement and a poor choice of an example, or whether a double Gaussian fit is indeed merited.

Please label the structures that resemble "rosettes".

Revision Plan

Good idea, which we will implement.

Lu Gan

Reviewer #2 (Significance (Required)):

General - This is a simulation-centric study of mammalian chromosome compaction that supports the loop-capture mechanism. It may be viewed as provocative by some readers because loop-extrusion has dominated the chromosome-compaction literature in the past decade. The only limitation, which is best addressed by future studies, is the absence of more direct molecular evidence of loop capture in situ. Though this same limitation applies to studies of the loop-extrusion mechanism.

Advance - It is valuable for the field to consider alternative mechanisms. In my opinion, the dominant one has been studied to death by indirect methods without a direct molecular-resolution readout in situ. While the field awaits better experimental tools, more mechanisms should be explored.

Audience - The chromosome-biology community (both bacterial and eukaryotic) will be interested.

Expertise - My lab uses cryo-ET to study chromatin in situ.

Reviewer #3 (Evidence, reproducibility and clarity (Required)):

In this manuscript, Kakui et al. measured the length/width relationships of mitotic chromosomes in human cells that had entered mitosis for different durations. This simple measurement revealed very interesting behaviors of mitotic chromosomes. They found that the longer chromosome arms were wider than shorter ones. Mitotic chromosomes became progressively wider over time, with shorter ones reached the final state faster than the longer ones. They then built a loop-capture polymer model, which explained the time-dependent increase of width/length ration rather well, but did not quite explain the final roundness of chromosomes. I suggest the following points for the authors to consider.

Major points

(1) There is no experimental evidence that the loop capture mechanism is condensin-dependent. Can the authors deplete condensin I or II or both and measure chromosome length and width in similar assays? This will link their models to molecular players.

Such analyses have been conducted by others, and we will provide a brief survey with relevant references to the literature in our revised introduction.

(2) It seems rather intuitive to me that if one defines the spacing the condensin-binding sites, then

Revision Plan

the loop sizes will be the same between shorter and longer chromosomes. It then follows that shorter chromosomes are rounder. Is it that simple? If not, can the authors provide a better explanation.

The reviewer makes an interesting point that roundness (width-length ratio), is greater for shorter chromosome arms, even if chromosome width is constant. We will make this clear in the revised manuscript.

(3) If the loop sizes are the same between shorter and longer chromosomes, why can't loop extrusion model explain this phenomenon? If one assumes that condensin is stopped by the same barrier element and has the same distribution at the loop base, this should produce the same outcome as loop capture.

The key feature of loop extrusion is the formation of a linear condensin backbone, resulting in a bottle brush-shaped chromosome. This arrangement prevents further equilibration of loops into a wider structure, as occurs in the loop capture mechanism by rosette rearrangements. These differences will be better explained, using a schematic, in the revised manuscript.

Minor points

(1) "We are aware that this approximation underestimates the length of the longest chromosome arms and overestimates the length of the shortest arms." should be "We are aware that this approximation underestimates the length of the longer chromosome (q) arms and overestimates the length of the shorter (p) arms.". Right?

In fact, this comparison applies to all longer and shorter arms, not only pairs of p and q arms, which we will clarify.

(2) Some scientists argue that the final chromosome conformation might be kinetically driven. Even if the short chromosomes have reached the final roundness, this doesn't necessarily mean that they have reached equilibrium in cells. "Steady state" might be a better term to describe the chromosomes in vivo, as there are clearly energy-burning processes.

The reviewer is right that the term 'equilibrium' can be seen as misleading, which we will replace with 'steady state'.

Reviewer #3 (Significance (Required)):

I find the paper intellectually stimulating and a pleasure to read. It suggests a plausible explanation for mitotic chromosome formation. As such, it will be of great interest to scientists in the chromatin field.

Reviewer #4 (Evidence, reproducibility and clarity (Required)):

The take home message of this study is that chromosome structure can be attained through mechanisms of looping that do not require an explicit loop extrusion function. As the authors

Revision Plan

states, alternative models of loop capture have been proposed, dating from 2015-2016. These models show DNA chains through simply Brownian diffusion can adopt a loop structure (citation 27, 28 and similarly Entropy gives rise to topologically associating domains Vasquez et al 2016 DOI: 10.1093/nar/gkw510).

The reviewer makes an excellent point in that entropy considerations, e.g. depletion attraction, likely contribute to the efficiency of loop capture. We will refer to this principle, including a citation to the Vasquez et al. study, in the revised manuscript.

In this study, the authors go through careful and well-documented chromosome length measurements through prophase and metaphase. The modeling studies clearly show that loop capture provides a tenable mechanism that accounts for the biological results. The results are clearly written and propose an important alternative narrative for the foundation of chromosome organization.

Reviewer #4 (Significance (Required)):

The study is important because it takes a reductionist approach using just Brownian motion and loop capture to ask how well the fundamental processes will recapitulate the biological outcome. The fact that loop capture can account for the arm length to width relationships on biological time scales is important to report to the community.

The work is extremely well done and the analysis of chromosome features is thorough and well-documented.

3. Description of the revisions that have already been incorporated in the transferred manuscript

We have begun work on the revisions as indicated above, but they have not yet been incorporated into the transferred manuscript.

4. Description of analyses that authors prefer not to carry out

We intend to address all the reviewers' comments.

Dear Dr. Uhlmann,

Thank you for transferring your manuscript to EMBO Reports, which was previously reviewed at Review Commons.

Referees express interest in your study analyzing chromosome shape changes during cell division and supporting a 'diffusion capture' model for chromosome condensation. However, they also raise concerns that need to be addressed to consider publication in EMBO Reports.

Having looked at all documents, we would like to invite you to submit a revised manuscript as in your revision plan. Please revise your manuscript with the understanding that the referee concerns (as in their reports) must be fully addressed and their suggestions taken on board. Please address all referee concerns in a complete point-by-point response. Acceptance of the manuscript will depend on a positive outcome of a second round of review. It is EMBO reports policy to allow a single round of major experimental revision only and acceptance or rejection of the manuscript will therefore depend on the completeness of your responses included in the next, final version of the manuscript.

We realize that it is difficult to revise to a specific deadline. In the interest of protecting the conceptual advance provided by the work, we recommend a revision within 3 months. Please discuss the revision progress ahead of this time with me if you require more time to complete the revisions, or if you have questions or comments regarding the revision (also by video chat).

1. A data availability section providing access to data deposited in public databases is missing (where applicable).
2. Your manuscript contains statistics and error bars based on $n=2$. Please use scatter plots in these cases.

You can submit the revision either as a Scientific Report or as a Research Article. For Scientific Reports, the revised manuscript can contain up to 5 main figures and 5 Expanded View figures, and it should not exceed 27000 characters. If the revision leads to a manuscript with more than 5 main figures it will be published as a Research Article. In this case the Results and Discussion section should be separate. If a Scientific Report is submitted, these sections have to be combined. This will help to shorten the manuscript text by eliminating some redundancy that is inevitable when discussing the same experiments twice. In either case, all materials and methods should be included in the main manuscript file.

4) a .docx formatted letter INCLUDING the reviewers' reports and your detailed point-by-point responses to their comments. As part of the EMBO publication's Transparent Editorial Process, EMBO reports publishes online a Review Process File (RPF) to accompany accepted manuscripts. This File will be published in conjunction with your paper and will include the referee reports, your point-by-point response and all pertinent correspondence relating to the manuscript.

<https://www.embopress.org/page/journal/14693178/authorguide#transparentprocess>

5) a complete author checklist, which you can download from our author guidelines <https://www.embopress.org/page/journal/14693178/authorguide>. Please insert information in the checklist that is also reflected in the manuscript. The completed author checklist will also be part of the RPF.

6) Please note that all corresponding authors are required to supply an ORCID ID for their name upon submission of a revised manuscript (). Please find instructions on how to link your ORCID ID to your account in our manuscript tracking system in our Author guidelines

Additional information on source data and instruction on how to label the files are available:
<https://www.embopress.org/page/journal/14693178/authorguide#sourcedata>

9) Our journal encourages inclusion of *data citations in the reference list* to directly cite datasets that were re-used and obtained from public databases. Data citations in the article text are distinct from normal bibliographical citations and should directly link to the database records from which the data can be accessed. In the main text, data citations are formatted as follows: "Data ref: Smith et al, 2001" or "Data ref: NCBI Sequence Read Archive PRJNA342805, 2017". In the Reference list, data citations must be labeled with "[DATASET]". A data reference must provide the database name, accession number/identifiers and a resolvable link to the landing page from which the data can be accessed at the end of the reference. Further instructions are available at <http://www.embopress.org/page/journal/14693178/authorguide#referencesformat>

12) Please also note our reference format:
<http://www.embopress.org/page/journal/14693178/authorguide#referencesformat>

13) All Materials and Methods need to be described in the main text using our 'Structured Methods' format, which is required for all research articles. According to this format, the Methods section includes a Reagents and Tools Table (listing key reagents, experimental models, software and relevant equipment and including their sources and relevant identifiers) followed by a Methods and Protocols section describing the methods using a step-by-step protocol format. The aim is to facilitate adoption of the methodologies across labs. More information on how to adhere to this format as well as a downloadable template (.docx) for the Reagents and Tools Table can be found in our author guidelines:
<https://www.embopress.org/page/journal/14693178/authorguide#structuredmethods>.

An example of a Method paper with Structured Methods can be found here:
<https://www.embopress.org/doi/10.15252/msb.20178071>.

I look forward to seeing a revised version of your manuscript when it is ready. Please let me know if you have questions or comments regarding the revision.

Kind regards,

Deniz Senyilmaz Tiebe

Deniz Senyilmaz Tiebe, PhD
Senior Scientific Editor
EMBO Reports

We would like to thank the four reviewers for their interest in our study, as well as for their constructive critique and suggestions for developing our investigations. We have performed a series of additional experiments and analyses, which we hope the reviewers will agree have substantially strengthened the conclusions of our manuscript. Please find in the following a point-by-point response to all the reviewers' comments.

Reviewer #1 (Evidence, reproducibility and clarity (Required)):

Summary: This manuscript authored by Kakui and colleagues aims to understand on how mitotic chromosomes get their characteristic, condensed X shape, which is functionally important to ensure faithful chromosome segregation and genome inheritance to both daughter cells. The authors focus on the condensin complex, a central player in chromosome condensation. They ask whether it condenses chromosomes through a now broadly popular "loop-extrusion" mechanism, in which a chromatin-bound condensin complex reels chromatin into loops until it dissociates or encounters a roadblock on the polymer (another condensin or some other protein complex), or through an alternative, "diffusion-capture" mechanism, in which a chromatin-bound condensin complex forms loops by encountering another chromatin-bound condensin until they dissociate from DNA (or from each other.)

The authors measured the progressive changes in the shape of mitotic chromosomes by taking samples at given time points from synchronized and mitotically arrested cells and found that while all chromosomes become more condensed and shorter, their width correlated with the length of the chromosome arms. They also observed that chromosome compaction/shortening evolves on a time scale much longer than the interval between the onset of chromosome condensation and the start of chromosome segregation, suggesting that chromatin condensation does not reach its steady-state during an unperturbed mitosis. The observed width-length correlation could be described by a power law with an exponent that increases with the time (i.e. chromosome condensation). The authors also performed polymer simulations of the diffusion-capture mechanism and found that the simulations semi-quantitatively recapitulate their experimental observations.

Major Comments

My most substantial comments focus on somewhat technical details of the image analysis approaches taken and the polymer models employed. However, as all reported data are derived from those details, I feel it is crucial to address them.

We thank the reviewer for the appraisal of our work, and for the suggestions on how to improve our image analysis and polymer modeling approaches. We have developed both aspects of our manuscript with additional experiments and analyses, as detailed below.

1. Definition/measurement of chromatin arms width and length. The approach taken to manually threshold an "arm" object and then fitting it with a same-area ellipse is not an ideal approach to gauge length and width of the arm, for the following reasons: (1) An ellipse appears to do a poor job approximating many of the objects that we see in the zoom-in insets of Fig. 1. Importantly, for somewhat bent shapes we see in the insets it likely strongly underestimates the length of the arms; this approach also presents potential problems for measuring width as well (see 2 and 3 here). (2) One concern is that, due to the diffraction limit, a cylindrical fluorescent object could appear somewhat wider at the mid-length than the real underlying cylinder or the poles; this effect could become more pronounced as the object gets brighter and shorter. (3)

Forcing the fit to an ellipse to objects that are not truly rod-shaped can drive an overestimation of the width of the object, and I suspect that this effect also might correlate with the length and brightness of the object. (4) Given 1-3 above, I think the approach the authors used for the first two time points, while not perfect, is better suited and likely more robust while avoiding these caveats. Moreover, why the authors cannot use this same approach (but just for each arm separately) for the later (30+ min) time points as they used for first two is unclear. This point is underscored by the observation that there is a drastic difference in the results between the first two and all subsequent points. When the authors compared the two approaches at the 30 min time point (where width-length dependence is still weak) in different cell lines they did indeed see different results (Fig. S2), although they concluded that the difference was acceptable.

As suggested by the reviewer, we have used the same approach that we used for the first two time points and applied a moving Gaussian fit to chromosome arms at all later time points. This approach is documented in the revised Figure EV2 and yielded qualitatively similar results as the ellipsoid fit data reported in Figure 2. In the subsequent figures, we analyze the measurements obtained from ellipsoid fits. The choice for this method is motivated by its ability to survey all arms, instead of just a subset, as well as by the observation that chromosome arms, especially shorter arms, visually resemble ellipsoids at later times.

Along these lines, the difference between short and long arms for the chromosome in the insets of Fig. 1 are quite subtle, except maybe at 180 and 240 min. On a related note, it might be informative to compare data for the two sister chromatid arms (as the underlying polymer has the same length) long vs long and short vs short and long vs short to help establish the robustness of the approach.

In the revised Figure 1, we selected insets that show chromosomes with a greater differential between short p- and long q-arm lengths. This choice makes the length-dependence of chromosome arm widths visually more apparent, not only when looking at the aggregate analysis of all chromosome arms, but also within each chromosome.

2. Regarding the power-law distribution, it is hard to judge based on the presented data whether it is a really good description of the data or not. In Fig. 1c, the points for a given time can barely be distinguished, while in Fig. 1b the authors plot individual time points in the panels, but the fits and points are overlapping so much that it is challenging to the main trends described by the clouds. The most informative approach for the reader would be to provide confidence intervals of the best fit parameters for all parameters that were varied in the fit. As the authors make some conclusions based on the power-law exponent values they observed, it would be helpful to know how confident we are in those values.

Confidence intervals of the power law fits are included in the revised Figure 2C. Additionally, to aid visual inspection of the data, we display the length to width relationships separated by timepoint in the new Figure EV2.

3. The conclusion that short arms equilibrate faster based on Fig. 3a is not fully convincing. For example, in a scenario where ~1.5 microns is the equilibrium length for all arms, and that the longest arms equilibrate the fastest - you would see the same qualitative pattern for quantiles, not much change in low percentiles, while you would observe a decrease in the values for the

high percentiles. The authors might be right, but Fig. 3A does not unambiguously demonstrate that it is so based on this evidence alone.

Our reasoning is based on the observation that the shortest percentiles do not change or do not change rapidly after 30 minutes, while the longest percentiles are clearly still relaxing towards a steady state. The observed pattern of all length percentiles is suggestive of graded asymptotic lengths. This said, the reviewer is right that the final steady state of longer arms was not captured in our analysis, and we have added a statement to this effect.

4. As for chromosome roundness, typically in image analysis, roundness is defined through the ratio of $(\text{perimeter})^2/\text{area}$; it might be better to use "aspect ratio" for the metrics used by the authors. And, perhaps, one should expect that shorter (measured, not necessarily by polymer contour length) arms should have a higher width/length ratio? If one selects for more round objects, there should be no surprise that the width and length get almost proportional. Given all of this, I am not sure whether width/aspect ratio serves as a good proxy for the chromatin condensation progression, which is how the authors are employing this data in the manuscript as written.

We thank the reviewer for pointing out that there is more than one definition of ‘roundness’. Definitions that we are aware of are based radii. E.g. the ISO definition of roundness is ‘the ratio of the inscribed to circumscribed radii’, equivalent to our ratio of the short and long chromosome arm axes. In the revised manuscript we better explain the roundness definition that we apply.

5. For the diffusion-capture model simulations, I think the results of the simulation would strongly depend on the assumptions of the probability to associate and the time scale of dissociation of the beads representing the condensin complex. For example, for a very strong association one might expect that all condensin will end up in one big condensate, even in the case of a long polymer. This is not explored/discussed at all. Did the authors optimize their model in any way? If not, how have they estimated the values they used? Moreover, perhaps this is an opportunity to learn/predict something about condensin properties, but the authors do not take advantage of this opportunity.

We explored a range of diffusion capture probabilities when we initially developed our loop capture simulations, and we found that the simulations were robust to parameter choice (Gerguri et al. 2021). During the revisions, we additionally tested a range of loop capture probabilities when simulating the behavior of various length chromatin chains. We altered the looping probability over two orders of magnitude and, again, found little impact on the outcome. These observations, documented in a new Figure EV4B, suggest that the observed polymer shape changes are limited by the diffusive behavior of the polymer, more so than by the dynamics of condensin’s tethering action.

In addition, the authors did some checks to show that the steady-state results of the simulations do not depend on the initial conditions. However, as some of the results reported concern the polymer evolution to the steady state (Fig. 6b-c), they also need to examine whether these results depend on the chosen initial conditions (or not), and if they do, what is the rationale for the choices the authors have made?

We compare steady states that are reached after starting our simulations from two distinct initial conditions. (i) a random walk, as an agnostic approach to investigate polymer behavior. (ii) because chromosome formation does not appear to initiate from a random walk inside cells, but rather from an elongated state, we modeled our second initial condition as a four times elongated state, approximating that observed in cells. The steady states reached in both simulations were indistinguishable, an observation that is documented in Figure EV4A.

6. A more thorough discussion of other possible models, beyond diffusion-capture model considered here, would be beneficial to the reader. First, the authors practically discard the possibility of the loop-extrusion model to explain their observations (although they never explicitly state this in the abstract or discussion). However, they neither leveraged simulations to rigorously compare models nor included some other substantiated arguments to explain why they prefer their model. This is important, as one of the major findings here is that the chromatin never reaches steady state for condensation, making it challenging to intuit what one should expect in this very dynamic state. Second, the authors, while briefly mentioning that there might be some other mechanisms contributing to the mitotic chromosome reshaping, do not really discuss those possibilities in a scholarly way. For example, work by the Kleckner group has suggested an involvement of bridges between sister chromatids into their shortening dynamics (Chu et al. Mol Cell 2020). Third, the authors do not discuss how they envision the interplay between the different SMC complexes - cohesin, condensin I and condensin II - as they act on the same chromatin polymer, or at least acknowledge a possible role that this interplay might contribute to the observed time dependencies.

We concede that the simulations presented in our manuscript are necessarily a simplification. Native chromosomes are bound by many more proteins additional to condensin. This being said, the purpose of the current manuscript is focused onto investigating the effect that condensin-mediated loop capture interactions exert on the behavior of a chromatin chain.

Despite the focus of our study, we agree with the reviewer on the importance of putting our model into biological context. In the revised manuscript we therefore (i) include a more thorough introduction and discussion of the loop extrusion mechanism and how it differs from loop capture, including references to previous studies that have modeled loop extrusion. (ii) we discuss additional mechanisms that contribute to chromosome formation, including sister links provided by cohesin (Chu et al. 2020), two types of condensin in most eukaryotes, as well as others.

Reviewer #1 (Significance (Required)):

Significance:

The question the authors are trying to address is fundamental and important. While loop extrusion-driven mitotic chromosome organization is a popular model, considering alternative models is always crucial, especially when one can find experimental observations that allow us to discriminate between possible models. The main limitations are: 1) the performance of the approach the authors take to measure chromosome shape is in question and 2) the main competitive model (loop extrusion) is not modeled. If all shortcomings are addressed this work may provide strong evidence for the diffusion-capture model and thus advance our mechanistic understanding of mitotic processes, which will be of broad interest to the fields of genome and chromosome biology.

During our revisions, we have introduced an additional approach to measure chromosome shape, as well as included a more comprehensive discussion of the loop extrusion mechanism. We hope the reviewer agrees that characterizing ‘shape change over time’ is an original and valuable approach that adds to our understanding of chromosome behavior.

No single approach will be able to capture all aspects of chromosome behavior, and we have strived to better place our observations into context in the revised manuscript.

Reviewer #2 (Evidence, reproducibility and clarity (Required)):

SUMMARY

The authors tracked the progression of mitotic chromosome compaction over time by imaging chromatin spreads from HeLa cells that were released from G2/M arrest. By measuring the mitotic chromosome arms' width and length at different times post-release, the authors demonstrated that the speed at which the chromosome arms reach an equilibrium state is dependent on their length. The authors were able to recapitulate this observation using polymer simulations that they previously developed, supporting the model of loop capture as the mechanism for mitotic chromosome compaction.

MAIN COMMENTS

This is a straightforward paper that supports an alternative mechanism (relative to the highly popular loop-extrusion) model for chromosome compaction. My comments are meant to help the manuscript reach a wider audience.

I suggest that "equilibrium" be replaced with "equilibrium length" since it is the only equilibrium parameter of concern.

The reviewer is right, we have implemented this change. We have also considered the reasoning of reviewer 3 that ‘steady state’ is a better term than ‘equilibrium’ to describe a shape that is maintained by an active process.

In the results, it may help to describe how loop capture and loop extrusion are incorporated into the simulations, using terminology that non-experts can understand. Such a description should be accompanied by figures that can be related to the other figures (color scheme, nomenclature if possible).

We now provide a better explanation of the loop capture and loop extrusion mechanisms in the revised introduction, and we return to their similarities and differences in the discussion. We have also included schematics in the revised Figures 4A and 5A that illustrate the behavior of a chromatin chain and the effect of loop capture.

OTHER COMMENTS

P5: Is it possible the chromosome-spread processing may distort the structures of the chromosomes?

The reviewer raises an important point when querying the impact of chromosome fixation and spreading. To explore this issue, we visualized chromosomes in live cells using the SiR-DNA stain. Because of crowding, we found it hard to trace entire chromosomes in live cells.

However, we can measure their widths. At both 12 and 20 minutes following G2 release, we found that chromosome widths develop quantitatively similar in live cells when compared to

our observations following fixation and spreading. These results suggest that sample processing did not unduly distort chromosome architecture. They are documented in a new Figure EV1C. We also note in our revised study that quantitatively similar chromosome widths in human prophase were recorded by Booth et al. 2016 using a yet different fixation protocol and serial block-face scanning electron microscopy.

Please clarify whether mitosis can complete after drug removal at the various treatment intervals.

To address this question, we performed an experiment similar to that shown in Figure 1. We synchronized cells in G2 using RO-3306, then released the cells and held them for six hours in a mitotic colcemid block. Finally, we washed out colcemid. Following this treatment, 85% of cells remained viable and completed cell division. These results indicate that cells and chromosomes remain in a physiologically viable state throughout the experiment. They are presented in a new Figure EV1B.

P6: "Our records are not, therefore, meant as an accurate absolute measure of individual arms. Rather, fitting allows us to sample all chromosome arms and deduce overall trends of chromosome shape changes over time"

It would be better to state this sentence earlier in this paragraph, or earlier in the section so that readers' expectations are curbed when they're reading the detailed analysis plan.

In response to a suggestion from reviewer 1, we employed a complementary image analysis tool to measure chromosome widths using moving Gaussian fits. This approach confirmed our observations of progressive chromosome length and width changes over time, and they are shown in the new Figure EV2. The paragraphs that report on both the ellipsoid and moving Gaussian fits have been rewritten.

P6: "As soon as individual chromosome arms become discernible (30 minutes), longer chromosome arms were wider, a trend that became more pronounced as time progressed." Implies that at early time points, when the lengths of the arms were unknown, the longer arms were equal or narrower than the short arms. I think it's more accurate to say that as soon as the arms were resolved, the longer arms appeared wider.

Thank you for this suggestion, which we adopted.

P7: Is there a functional consequence to the long arms not equilibrating before anaphase onset?

The reviewer raises an interesting question, which we explore at the end of our revised discussion. A consequence of not reaching 'steady state' is that 'time in mitosis' becomes a key parameter that defines chromosome arm length at anaphase onset.

P13: "In a loop capture scenario, we can envision how condensin II sets up a coarse rosette architecture, with condensin I inserting a layer of finer-grained rosettes." This should be illustrated in a figure.

The involvement of two condensin complexes in many species is intriguing, though their differential contributions to chromosome architecture remain to be more thoroughly explored in future experimental and simulation studies. We feel that our hypothesis of layered rosette

structures remains currently too speculative. Instead of illustrating this proposal, we have toned down its presentation.

FIGURES

Fig. 1: "...while insets show chromosomes at increasing magnification over time" sounds like the microscope magnification is changing over time. Please change "magnification" to "enlargement". Alternatively, if the goal of the figure is to illustrate the shape/dimensions change of the chromosomes over time, wouldn't it be better to keep all the enlargements at the same scale?

Good point, we have adjusted the wording as the reviewer suggests. In response to the reviewer's suggestion, we tried different enlargement ratios for the insets, but found that the increasing enlargement provides the best illustration of the differential width increase of short and long chromosome arms.

Fig. 2a plot: Does the distribution of normalized intensities really justify a Gaussian fit? I see a double Gaussian.

The (randomly) chosen example line scan in the original Figure would indeed be better described by a double Gaussian. However, this example was a rare incidence where sister chromatids appeared to begin splitting. Most chromosome width scans at the 12 minutes time point are best describe by a single Gaussian fit. We show a more representative example in the revised Figure 2A.

Please label the structures that resemble "rosettes".

Good idea, labels have been added to Figure 5.

Lu Gan

Reviewer #2 (Significance (Required)):

General - This is a simulation-centric study of mammalian chromosome compaction that supports the loop-capture mechanism. It may be viewed as provocative by some readers because loop-extrusion has dominated the chromosome-compaction literature in the past decade. The only limitation, which is best addressed by future studies, is the absence of more direct molecular evidence of loop capture in situ. Though this same limitation applies to studies of the loop-extrusion mechanism.

Thank you for these constructive comments on our manuscript. We agree that *in situ* evidence for how condensin shapes chromosomes remains a key target for future exploration. We are actively working on visualizing live chromosome formation both *in vitro* and *in vivo*, approaches that however require further methods development. We hope to report on *in situ* observations of condensin action during chromosome formation in the future.

Advance - It is valuable for the field to consider alternative mechanisms. In my opinion, the dominant one has been studied to death by indirect methods without a direct molecular-resolution readout in situ. While the field awaits better experimental tools, more mechanisms should be explored.

Audience - The chromosome-biology community (both bacterial and eukaryotic) will be interested.

Expertise - My lab uses cryo-ET to study chromatin in situ.

Reviewer #3 (Evidence, reproducibility and clarity (Required)):

In this manuscript, Kakui et al. measured the length/width relationships of mitotic chromosomes in human cells that had entered mitosis for different durations. This simple measurement revealed very interesting behaviors of mitotic chromosomes. They found that the longer chromosome arms were wider than shorter ones. Mitotic chromosomes became progressively wider over time, with shorter ones reached the final state faster than the longer ones. They then built a loop-capture polymer model, which explained the time-dependent increase of width/length ration rather well, but did not quite explain the final roundness of chromosomes. I suggest the following points for the authors to consider.

Major points

(1) There is no experimental evidence that the loop capture mechanism is condensin-dependent. Can the authors deplete condensin I or II or both and measure chromosome length and width in similar assays? This will link their models to molecular players.

The consequences of condensin depletion on chromosome architecture have been previously documented by us and others. Depending on depletion efficiency, mitotic chromosome formation is severely compromised or entirely abolished. We have included a statement in the revised introduction and give references to review articles that survey previous studies in different model organisms.

CHIA-PET and Hi-C analyses in the fission yeast model have furthermore illustrated that condensin engages in DNA capture interactions within mitotic chromosomes (Tang et al. 2023). Recent observations in vertebrate cells are also suggestive of a condensin capture mechanisms, even though they are not always portrayed in this way. In the revised introduction we refer to two of the most notable recent studies by Samejima et al, 2025 and by Beckwith et al, 2025.

(2) It seems rather intuitive to me that if one defines the spacing the condensin-binding sites, then the loop sizes will be the same between shorter and longer chromosomes. It then follows that shorter chromosomes are rounder. Is it that simple? If not, can the authors provide a better explanation.

The reviewer makes an important observation that chromosome roundness will be greater for shorter chromosome arms, even if chromosome width is constant. However, we observe that longer chromosome arms are wider than shorter arms (a conclusion that was corroborated during the revisions by additional moving Gaussian fit measurements). Importantly, this length-width relationship arises in spite of constant condensin spacing along long and short arms. Our simulations suggest an explanation for this behavior: loop rosettes of constant loop size arrange themselves according to the principles of polymer physics. This relationship is now better illustrated in the revised Figure 5.

(3) If the loop sizes are the same between shorter and longer chromosomes, why can't loop extrusion model explain this phenomenon? If one assumes that condensin is stopped by the same barrier element and has the same distribution at the loop base, this should produce the same outcome as loop capture.

The reviewer correctly points out that loop extrusion predicts the emergence of consecutively ordered loops, which emanate from a central condensin backbone in a bottle brush arrangement (Goloborodko et al. 2016). Given constant loop sizes, the loop extrusion model predicts that longer chromosomes are longer, but not wider. This prediction is better explained in our revised introduction, and it is contrary to our observations.

Minor points

(1) "We are aware that this approximation underestimates the length of the longest chromosome arms and overestimates the length of the shortest arms." should be "We are aware that this approximation underestimates the length of the longer chromosome (q) arms and overestimates the length of the shorter (p) arms.". Right?

Correct, thank you, we have amended the text as suggested.

(2) Some scientists argue that the final chromosome conformation might be kinetically driven. Even if the short chromosomes have reached the final roundness, this doesn't necessarily mean that they have reached equilibrium in cells. "Steady state" might be a better term to describe the chromosomes in vivo, as there are clearly energy-burning processes.

The reviewer is again right, chromosomes maintain their shape in an active process that requires continuous ATP hydrolysis by condensin (e.g. Kinoshita et al. 2015). 'Steady state' is the correct phrase to describe the final chromosome shape, and we have amended the manuscript accordingly.

Reviewer #3 (Significance (Required)):

I find the paper intellectually stimulating and a pleasure to read. It suggests a plausible explanation for mitotic chromosome formation. As such, it will be of great interest to scientists in the chromatin field.

Reviewer #4 (Evidence, reproducibility and clarity (Required)):

The take home message of this study is that chromosome structure can be attained through mechanisms of looping that do not require an explicit loop extrusion function. As the authors states, alternative models of loop capture have been proposed, dating from 2015-2016. These models show DNA chains through simply Brownian diffusion can adopt a loop structure (citation 27, 28 and similarly Entropy gives rise to topologically associating domains Vasquez et al 2016 DOI: 10.1093/nar/gkw510).

The reviewer makes an excellent point that entropy considerations, which in other contexts have been called 'depletion attraction', are instrumental for chromosome (domain) formation. We refer to this principle and include a citation to the key Vasquez et al. 2016 paper in the revised manuscript.

In this study, the authors go through careful and well-documented chromosome length measurements through prophase and metaphase. The modeling studies clearly show that loop capture provides a tenable mechanism that accounts for the biological results. The results are clearly written and propose an important alternative narrative for the foundation of chromosome organization.

Reviewer #4 (Significance (Required)):

The study is important because it takes a reductionist approach using just Brownian motion and loop capture to ask how well the fundamental processes will recapitulate the biological outcome. The fact that loop capture can account for the arm length to width relationships on biological time scales is important to report to the community.

The work is extremely well done and the analysis of chromosome features is thorough and well-documented.

We thank the reviewer for the positive appraisal of our work.

Dear Frank,

Thank you for submitting your revised manuscript. It has now been seen by both of the original referees.

As you will see, referees find that the study is significantly improved during revision and recommend publication. However, referee #1 has a few outstanding concerns, which are directly related to his/her initial criticisms. Please address these concerns and provide a point-by-point response. Please contact me if you would like to discuss any of these points further.

Moreover, the editorial points below need to be addressed before I can accept the manuscript.

- Please reduce the number of keywords to 5.
- The Acknowledgments and Disclosure and competing interests statement sections need to be moved after the Acknowledgements section.
- Please remove the Author Contributions section from the manuscript text.
- As per our format requirements, in the reference list, citations should be listed in alphabetical order and then chronologically, with the authors' surnames and initials inverted; where there are more than 10 authors on a paper, 10 will be listed, followed by 'et al.'. Please see <https://www.embopress.org/page/journal/14693178/authorguide#referencesformat>
- Funding information needs to be complete in both the manuscript text and the manuscript tracking system as per EMBO Press policy. We note that the following is currently missing in the manuscript tracking system: Dr. Yoshifumi Jigami Memorial Fund, The Society of Yeast Scientists, the Institute for Fermentation, Osaka (IFO) and Waseda University grants for Special Research Projects (2021C-387, 2022C-306, 2023C-283, 2024C-285, to Y.Ka.)
- Our production/data editors have asked you to clarify several points in the figure legends - Figure Legends (main + EV):
 - o Please note that the box plots need to be defined in terms of minima, maxima, centre, bounds of box and whiskers, and percentile in the legends of figures EV1 C
 - o Please note that the error bars are not defined in the legends of figures EV1 A, B
 - o Please note that the scale bar needs to be defined for figure EV2 B
- Papers published in EMBO Reports include a 'synopsis' and 'bullet points' to further enhance discoverability. Both are displayed on the html version of the paper and are freely accessible to all readers. The synopsis includes a short standfirst summarizing the study in 1 or 2 sentences (max 35 words) that summarize the paper and are provided by the authors and streamlined by the handling editor. I would therefore ask you to include your synopsis blurb and 3-5 bullet points listing the key experimental findings.
- In addition, please provide an image for the synopsis. This image should provide a rapid overview of the question addressed in the study but still needs to be kept fairly modest since the image size cannot exceed 550 (width) x 300-600 (height) pixels.

Thank you again for giving us to consider your manuscript for EMBO Reports, I look forward to your minor revision.

Kind regards,

Deniz

--

Deniz Senyilmaz Tiebe, PhD
Senior Scientific Editor
EMBO Reports

Referee #1:

Summary

The work by Kakui and colleagues reports measurements of mitotic chromosomes shape during progressive condensation under mitotic arrest. The authors report that while chromosome length shortens and their width increases, the width of chromosome arms correlates with their length (the authors report power law scaling), and that on a typical mitosis time scale the condensing chromosomes do not reach the steady state length and width. The authors also consider and simulate a diffusion-capture model in which the condensation is driven by condensin binding to the chromatin followed by loop formation through condensin-condensin attraction. The model simulations recapitulate some reported observations, making it a plausible explanation for the mechanism of mitotic chromosome condensation, although a major concern regarding the reported data and analysis compromises the strength of this evidence (see below).

Major Comment:

1. The manuscript seeks to provide evidence for a "diffusion-capture" mechanism for mitotic chromosomes condensation, which is an alternative to a more popular "loop-extrusion" mechanism. While we feel very enthusiastic about considering alternative mechanisms and about attempts to provide quantitative data that any mechanism/model should account for, we remain

concerns that the reported quantitative observations might be affected by the analysis approach. Namely, the authors measure the width and length of the chromosome arms by, first, segmenting the image with intensity thresholding and, then, fitting an ellipsoid to the segmented area. The potential caveat of this approach is that, due to the diffraction limit, any rod -shape (i.e. constant width) object will start to look "ellipsoidal" (i.e. with a wider "waist" in the middle) with the effect depending on the intensity and, hence, the length of the rod. The authors tried to address this concern, which we raised in the previous round of review, by performing a more manual labor-intensive analysis where they manually traced chromosome contours and fit a Gaussian to the intensity distribution along the lines orthogonal to the chromosome contour. The results are now presented in Fig. EV2 (thank you!), and the authors concluded that:

"This approach leads to a more accurate measurement of arm widths, but could not be applied to all chromosome arms, when they were curved or when sister chromatids were insufficiently separated. Despite these limitations, both the ellipsoid and moving Gaussian fits revealed a qualitatively similar development of chromosome dimensions from 30 to 360 minutes (Fig. EV2)"

However, at least in our opinion, Fig. EV2 actually reveals an importantly distinct pattern for the "Gaussian-fit" approach. In the Gaussian fit plot (right side) there are two "modes": for smaller chromosome arms there is a regime of roughly "linear" dependence (in alignment with a power law, given that plots are in logarithmic coordinates), while most arms display a width that is independent of their length in the regime of 0.8 -5 μ m. Moreover, the rationale provided by the authors for applying the ellipsoid-fitting approach to accommodate curved or not spatially separated chromosome arms is not compelling as we can expect that this approach would be even more prone to the systematic error described above in these scenarios.

Given that the key experimental results (and comparison to the model) in the manuscript hinge on the reported length and width of the chromosomes and on a "quantitative" relationship between width and length, a more thorough analysis is required to boost our confidence in the reported results.

Minor Comments:

1. We previously requested that the authors provide confidence intervals for each fitting parameter in the power-law fit (Response #2), as the fitting parameters might not be fully-independent resulting in poor confidence of individual parameters while providing a qualitatively close fit to the data. Since the authors use the scaling exponent as a quantitative argument to support their model, it remains important for them to demonstrate the degree of confidence in the reported values.
2. While the kinetics of the chain length change appears uncoupled from the probability of condensin-condensin interactions (as an aside, do the authors have an explanation as to why? Is it because the earlier stage kinetics are limited by the slow polymer during this phase?), exploring how the results depend on the lifetime (i.e. dissociation) of the condensin-condensin associations would potentially be more insightful (Response #4).
3. On a related note, the authors did not address the question of the potential influence of the initial conditions on the kinetics of the condensation (not on the steady-state (Response #5)). This question is not only important from a modeling perspective, but also for the model-experiment comparison as it is possible that in the experiment the initial state is different from the well-stretched state considered in the simulations.

Referee #2:

The revised manuscript is easier to read and will be a valuable contribution to the chromosome biology field. Congratulations!

Referee #1:*Summary*

The work by Kakui and colleagues reports measurements of mitotic chromosomes shape during progressive condensation under mitotic arrest. The authors report that while chromosome length shortens and their width increases, the width of chromosome arms correlates with their length (the authors report power law scaling), and that on a typical mitosis time scale the condensing chromosomes do not reach the steady state length and width. The authors also consider and simulate a diffusion-capture model in which the condensation is driven by condensin binding to the chromatin followed by loop formation through condensin-condensin attraction. The model simulations recapitulate some reported observations, making it a plausible explanation for the mechanism of mitotic chromosome condensation, although a major concern regarding the reported data and analysis compromises the strength of this evidence (see below).

Major Comment:

1. The manuscript seeks to provide evidence for a "diffusion-capture" mechanism for mitotic chromosomes condensation, which is an alternative to a more popular "loop-extrusion" mechanism. While we feel very enthusiastic about considering alternative mechanisms and about attempts to provide quantitative data that any mechanism/model should account for, we remain concerned that the reported quantitative observations might be affected by the analysis approach. Namely, the authors measure the width and length of the chromosome arms by, first, segmenting the image with intensity thresholding and, then, fitting an ellipsoid to the segmented area. The potential caveat of this approach is that, due to the diffraction limit, any rod-shape (i.e. constant width) object will start to look "ellipsoidal" (i.e. with a wider "waist" in the middle) with the effect depending on the intensity and, hence, the length of the rod. The authors tried to address this concern, which we raised in the previous round of review, by performing a more manual labor-intensive analysis where they manually traced chromosome contours and fit a Gaussian to the intensity distribution along the lines orthogonal to the chromosome contour. The results are now presented in Fig. EV2 (thank you!), and the authors concluded that: "This approach leads to a more accurate measurement of arm widths, but could not be applied to all chromosome arms, when they were curved or when sister chromatids were insufficiently separated. Despite these limitations, both the ellipsoid and moving Gaussian fits revealed a qualitatively similar development of chromosome dimensions from 30 to 360 minutes (Fig. EV2)" However, at least in our opinion, Fig. EV2 actually reveals an importantly distinct pattern for the "Gaussian-fit" approach. In the Gaussian fit plot (right side) there are two "modes": for smaller chromosome arms there is a regime of roughly "linear" dependence (in alignment with a power law, given that plots are in logarithmic coordinates), while most arms display a width that is independent of their length in the regime of 0.8 -5 μm . Moreover, the rationale provided by the authors for applying the ellipsoid-fitting approach to accommodate curved or not spatially separated chromosome arms is not compelling as we can expect that this approach would be even more prone to the systematic error described above in these scenarios. Given that the key experimental results (and comparison to the model) in the manuscript hinge on the reported length and width of the chromosomes and on a "quantitative" relationship between width and length, a more thorough analysis is required to boost our confidence in the reported results.

We appreciate the reviewer's additional feedback on the new results included in our revised manuscript, utilising Gaussian fitting to measure chromosome arm widths also at later times.

Firstly, we would like to emphasise that the ellipsoid and Gaussian fitting approaches agree with each other in revealing gradual chromosome shortening and widening over time. They also both suggest that longer arms become progressively wider than shorter arms.

At the same time, we agree with the reviewer that the Gaussian fits make the suggestion that the width plateaus with respect to length, for the longest arms. A plateau width of the longest arms is compatible with the idea that these arms widen at a uniform rate and are still on course to approach, but have not yet reached, steady state. However, the incomplete sampling makes it difficult to quantitatively compare the two sets of measurements, thereby preventing a firm conclusion. We now describe these observations in the EV2 figure legend.

Furthermore, we adapted the 'Measuring chromosome arm lengths and widths' section to better explain the benefits and limitations of both respective measurement approaches and, given the limitations, have removed the term 'quantified' from the article title.

Minor Comments:

1. We previously requested that the authors provide confidence intervals for each fitting parameter in the power-law fit (Response #2), as the fitting parameters might not be fully-independent resulting in poor confidence of individual parameters while providing a qualitatively close fit to the data. Since the authors use the scaling exponent as a quantitative argument to support their model, it remains important for them to demonstrate the degree of confidence in the reported values.

During the revisions, as rightly suggested by the reviewer, we have included confidence intervals with the power-law fits. These are shown in the revised Figure 2C.

2. While the kinetics of the chain length change appears uncoupled from the probability of condensin-condensin interactions (as an aside, do the authors have an explanation as to why? Is it because the earlier stage kinetics are limited by the slow polymer during this phase?),

We agree with the reviewer's interpretation, which we portrayed in the submitted revision, and which we now explain more clearly when referring to Fig. EV4B.

exploring how the results depend on the lifetime (i.e. dissociation) of the condensin-condensin associations would potentially be more insightful (Response #4).

We also agree that it would be interesting to explore condensin's association and dissociation rates separately. In the current simulations, the association probability is a single model parameter that controls both association and lifetime. To dissect the two reactions will be an interesting approach for a future study.

3. On a related note, the authors did not address the question of the potential influence of the initial conditions on the kinetics of the condensation (not on the steady-state (Response #5). This question is not only important from a modeling perspective, but also for the model-experiment comparison as it is possible that in the experiment the initial state is different from the well-stretched state considered in the simulations.

Indeed, Figure EV4A reports on the steady state, not on the kinetics of reaching steady state. When we initiate simulations from a random walk of steady state-like dimensions, there is very little change over time, making a kinetic analysis less informative.

Referee #2:

The revised manuscript is easier to read and will be a valuable contribution to the chromosome biology field. Congratulations!

Thank you.

Dr. Frank Uhlmann
The Francis Crick Institute
Chromosome Segregation Laboratory
1 Midland Road
London NW1 1AT
United Kingdom

Dear Frank,

Thank you for submitting your revised manuscript. I have now looked at everything and all is fine. Therefore, I am very pleased to accept your manuscript for publication in EMBO Reports.

Congratulations on a nice work!

Kind regards,

Deniz

Deniz Senyilmaz Tiebe

--

Deniz Senyilmaz Tiebe, PhD
Senior Scientific Editor
EMBO Reports

--
